# Reciprocal regulation between the molecular clock and kidney injury

Carlos Rey-Serra[1], Jessica Tituaña[1], Terry Lin[2], J Ignacio Herrero[1], Verónica Miguel[1], Coral Barbas[3], Anna Meseguer[4], Ricardo Ramos[5], Amandine Chaix[2], Satchidananda Panda[2], Santiago Lamas[1]

**Tubulointerstitial fibrosis is the common pathological substrate for many etiologies leading to chronic kidney disease. Although perturbations in the circadian rhythm have been associated with renal disease, the role of the molecular clock in the pathogenesis of fibrosis remains incompletely understood. We investigated the relationship between the molecular clock and renal damage in experimental models of injury and fibrosis (unilateral ureteral obstruction, folic acid, and adenine nephrotoxicity), using genetically modified mice with selective deficiencies of the clock components *Bmal1*, *Clock*, and *Cry*. We found that the molecular clock pathway was enriched in damaged tubular epithelial cells with marked metabolic alterations. In human tubular epithelial cells, TGFβ significantly altered the expression of clock components. Although *Clock* played a role in the macrophage-mediated inflammatory response, the combined absence of *Cry1* and *Cry2* was critical for the recruitment of neutrophils, correlating with a worsening of fibrosis and with a major shift in the expression of metabolism-related genes. These results support that renal damage disrupts the kidney peripheral molecular clock, which in turn promotes metabolic derangement linked to inflammatory and fibrotic responses.**

## Introduction

Organ fibrosis is an unfavorable consequence of chronic inflammation characterized by an excessive accumulation of ECM. In the kidney, fibrosis is the ultimate stage of the cellular response to chronic inflammation and the converging pathological substrate for several entities of different origins, leading to chronic kidney disease (CKD) [1]. Whereas the understanding of the molecular and cellular basis of organ fibrosis, including that of the kidney, has experienced substantial progress in the last decade ([2], [3], [4]), options for its prevention or treatment are scarce, with only a few drugs

offering limited therapeutic advantage in the case of idiopathic pulmonary fibrosis [5]. Renal tubulo-interstitial fibrosis is characterized by a profound alteration of metabolism within the tubular epithelial cell (TEC), majorly related to a drastic reduction in fatty acid oxidation (FAO) [6]. We recently showed that genetically mediated FAO enhancement in the renal tubule resulted in significant protection from fibrosis in several experimental models [7]. This and other studies support an important role for metabolic failure in kidney fibrosis, whereby inflammation and mitochondrial dysfunction contribute to perpetuate a vicious cycle that results in ECM deposition, fibrosis, and progression of CKD.

Many physiological functions of most tissues and cells across living organisms exhibit daily periodic fluctuations related to the circadian rhythm, majorly conditioned by predictable environmental cues such as the light/dark cycle. The circadian rhythm is composed by a master clock, located in the suprachiasmatic nucleus of the hypothalamus, which is entrained by light and can synchronize tissue and cell peripheral clocks via neuronal and humoral signals [8]. At the molecular level, the peripheral clocks are regulated by a transcriptional feedback loop in which clock components such as *Arntl1* (Bmal1) and *Clock* activate the transcription of their own repressors including *Per* and *Cry* [9]. Perturbations in the circadian rhythm, including the molecular clock, have been associated with many pathologies including renal disease ([10], [11], [12], [13], [14]). In addition, there is an important crosstalk between the circadian clock and metabolism [15]. Nutrient flux in the bloodstream fluctuates considerably throughout the day as a consequence of diurnal behavioral rhythmicity. Consequently, the levels of most metabolites, including glucose, amino acids, and lipids oscillate in the blood in a synchronous manner with the environmental time. In the kidney, the circadian clock modulates blood flow, glomerular filtration rate, and ion and water excretion. However, the bidirectional influence between fibrosis and alterations of the molecular clock remains incompletely understood. By using several models of kidney injury in the setting of mice genetically modified for key components of the molecular clock, we aimed to clarify this question. In addition, we

---

[1]Program of Physiological and Pathological Processes, Centro de Biología Molecular Severo Ochoa (CSIC-UAM), Madrid, Spain    [2]Regulatory Biology Laboratory, Salk Institute for Biological Studies, La Jolla, CA, USA    [3]Centre for Metabolomics and Bioanalysis (CEMBIO), Department of Chemistry and Biochemistry, Facultad de Farmacia, Universidad San Pablo-CEU, Madrid, Spain    [4]Renal Physiopathology Group, Vall d'Hebron Research Institute (VHIR)-CIBBIM Nanomedicine, Barcelona, Spain    [5]Genomic Facility, Fundación Parque Científico de Madrid, Madrid, Spain

Correspondence: slamas@cbm.csic.es

have perused Sc-RNAseq data bases from studies in models akin to those presented here to analyze the transcriptional profiles of circadian-related genes. We found that unilateral ureteral obstruction (UUO)-induced kidney damage was associated with alterations in the molecular clock in immune cells and in PT cells with a shift in the gene expression profile of metabolism-related genes. We further observed that TGFβ1 increases the expression of molecular clock genes in human TECs. We found that selective deficiencies in the core clock components *Bmal1*, *Clock*, and *Cry* were associated with differential responses regarding inflammation and fibrosis. Strikingly, the absence of both Cry proteins 1 and 2 promoted a major change in the expression of metabolism-related genes, accompanied by neutrophil infiltration and a marked aggravation of the fibrotic phenotype.

# Results

### The molecular clock pathway is enriched in injured PT cells after kidney damage

To investigate the influence of kidney fibrosis on the expression of core clock components of the circadian machinery, we determined the gene expression levels of different molecular clock genes in mice subjected to UUO for 3, 7, 15, or 25 d (Fig 1A). We found that after UUO, the expression of *Arntl* was increased at all time points analyzed in comparison with their contralateral kidneys. Interestingly, its expression progressively increased, peaking 25 d after UUO (Fig 1B). The analysis of the mRNA levels also revealed differences in the expression of other clock genes (Fig 1B). Spearman correlation analysis revealed a significant positive correlation between several molecular clock genes and the mRNA levels of the fibrosis-related genes *Tgfb1* and *Fn1* after UUO (Figs 1C and S1A and Table S1). These data spotted *Arntl* and *Arntl2* as the molecular clock genes with the highest correlation with *Tgfb1* (0.71 and 0.88, respectively) (Fig 1C). This pattern was also recapitulated in two other models of kidney injury, resulting in an increase of its expression at 7 and 15 d after folic acid nephropathy (FAN) and at 25 d after adenine (ADN) (Fig S1B–E). Of note, the UUO model presented higher differences in the expression levels of *Arntl* compared with the FAN and ADN models for the same time point, a pattern also detectable with *Tgfb1, Fn1*, and collagen deposition (Figs 1B and S1B, C, and F–H). Moreover, the UUO model was associated with significantly more collagen deposition than the FAN and ADN models; 15 d after FAN and 25 d after ADN showed a deposition of collagen equivalent to 3 and 7 d after UUO, respectively. This set of observations, together with the greater variability observed in FAN and ADN due to a higher mortality rate associated to these models, pointed to the UUO as the most reliable model for further analysis. Thus, our rationale for the timing of the tissue collection in the different models was based on the differences in the progression of tissue damage present in the models analyzed. To develop a comprehensive overview of the changes associated to the molecular clock in kidney fibrosis, we analyzed the scRNA-seq database available from a study performed under comparable experimental conditions to our study, which included kidney samples from similar age and mouse genotype

subjected to sham or UUO for 7 d (16) (Fig 1D). The sample collection of this study took place between ZT2 and ZT5 under a 12/12 light/dark cycle (Susztak K, personal communication). For the analysis, unbiased clustering identified 22 populations that were annotated based on previously reported marker gene expression (16) (Fig 1E). We identified an increase in the expression of molecular clock genes in kidneys subjected to UUO compared with sham kidneys (Fig 1F). KEGG analysis revealed that the circadian rhythm pathway was enriched in immune cell–related clusters from UUO samples compared with sham kidneys. This finding was also consistent with enrichment in pathways related to the activation of the immune response (Fig 1F and G). However, the expression of the molecular clock genes was not significantly higher in the immune cells from UUO compared with sham (Fig 1H). As expected, the clusters related to the immune cells were mostly composed by cells from UUO (ranging from 85–100%) and, in consistence, most of the cells with positive expression of molecular clock genes came from the UUO kidneys (Fig 1I). These data suggest that the enrichment of the circadian pathway in UUO compared with sham kidneys observed by KEGG analysis (Fig 1G) is presumably due to the larger presence of immune cells in UUO samples and that most likely there is covariation in the expression of circadian and immune response–related genes.

The expression of several molecular clock genes was up-regulated in a subcluster of proximal tubule epithelial cells (PT) in kidneys subjected to UUO (Fig 1J and K), in good correlation with the enrichment of the "KEGG-circadian-rhythm-pathway" (Fig 1L). For the sake of simplicity, we named this subcluster "PT circadian." PT circadian cells were positive for the well-known PT cell markers Lrp2, Slc34a1, and Slc13a3 (16) (Fig 1M), confirming that these cells are PT cells. Gene Ontology analysis performed with DAVID, comparing the differential gene expression in both subclusters of PT cells from UUO kidneys, revealed a down-regulation of genes implicated in mitochondrial ATP synthesis and translation in PT circadian compared with PT cells, whereas the up-regulated genes were implicated in transcription, chromatin organization, RNA splicing and processing, cellular response to DNA damage, and cell migration, among others (Fig 1N). To make sure these changes could be generalized to other models of kidney damage, we analyzed the SC database of healthy mouse kidney and of kidneys at days 2 and 30 after ischemia-reperfusion injury (IRI) model (17). As shown in Fig S2, we identified enrichment of the "KEGG-circadian-rhythm-pathway" 30 d after IRI that was not detected in healthy kidneys or 2 d after IRI (Fig S2A). Furthermore, we identified a similar "PT circadian" subcluster within a PT cluster in the 30-d time point (Fig S2B–D). We also found increased expression of Bmal1, Clock, and Per and a pattern of variation of gene expression in the biochemical and cellular pathways similar to the UUO model (Fig S2E). Overall, these data suggest that the expression of molecular clock genes in PT cells after kidney damage correlates with metabolic impairment signatures.

### TGFβ up-regulates the expression of molecular clock genes in human proximal TECs (HPTEC)

TGFβ, the archetypal profibrotic cytokine is significantly involved in kidney fibrosis in humans (18). Thus, we evaluated its effect on

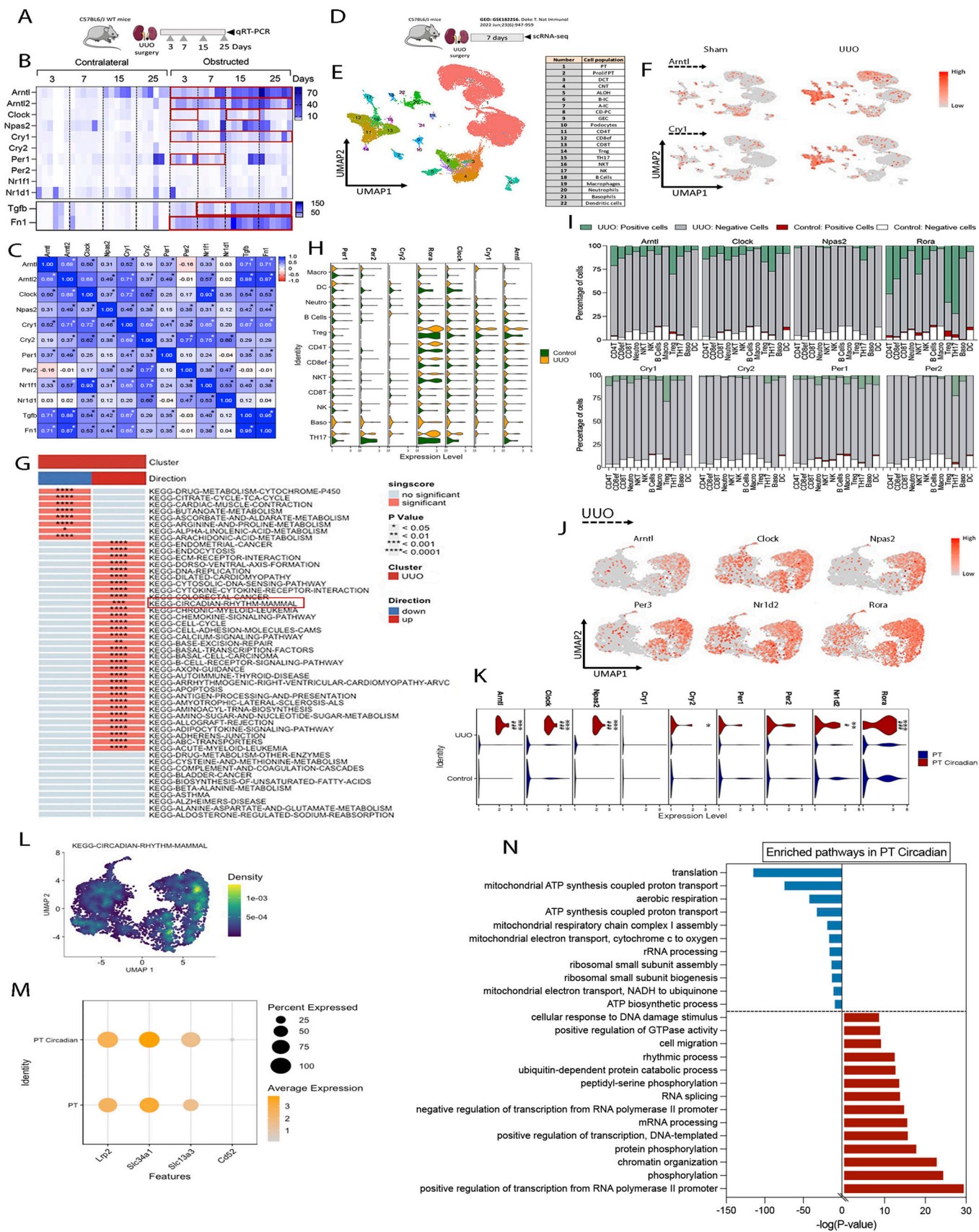

the expression of clock components in an immortalized cell line of HPTEC. As expected, TGFβ increased the expression of the fibrotic markers *FN1* and *COL1A1* in HPTEC (Fig 2A). Interestingly, treatment of HPTEC with TGFβ augmented the expression of *ARNTL, ARNTL2, CLOCK,* and *CRY1* in a time-dependent fashion (Fig 2A–C). The protein expression of Smad3 significantly increased after treatment with 2.5 µM SB505124, a TGFβ receptor antagonist (Fig 2D and E). The latter observation could represent a compensatory response as a consequence of the inhibitory effects of the TGFβ pathway antagonist. TGFβ-mediated increased in *ARNTL*/Bmal1, and *CRY1* expression was significantly reduced by the treatment with the inhibitor (Fig 2D–H), thus attesting to the mediation of TGFβ1 signaling. Of note, cells treated with SB505124 showed no significant alteration in terms of viability at any of the concentrations tested (Fig S3A). The data in HPTEC cells correlate with the results obtained in primary kidney cells from C57BL6/J WT mice (Fig S3B) and in human primary PTEC (HprimPTEC) (Fig S3C–E). Moreover, our data revealed that the *ARNTL* gene is the clock component showing the earliest up-regulation, increasing just 4 h after TGFβ1 treatment (Fig 2A). Furthermore, the analysis of *Arntl* pre-mRNA levels revealed an increase in their expression in fibrotic kidneys 25 d after UUO and in HprimPTEC after TGFβ1 treatment (Fig 2I–K), suggesting a potential transcriptional mechanism. In silico evaluation of the Bmal1 gene, *ARNTL*, using the R package for JASPAR2018 (20), revealed two potential binding sites for Smad3/4 with a relative score of 88/96% and 83%, respectively (Fig 2L). These sites belong to a distal enhancer located at 8,376 bp upstream from the transcription start site (Fig 2M) and are highly conserved among vertebrates (Fig 2N). However, overexpression of Smad3 did not result in an increase of *ARNTL* expression (Fig 2O–Q). In keeping, the analysis of bioluminescence reporter assays in the presence of plasmids bearing the enhancer regions corresponding to the Smad3-binding sites described above did not result in a significant increase after Smad3 overexpression (Fig 2R). Although this is compatible with a Smad3-independent regulation of Bmal1, the fact that the expression of *COL1A1* and the phosphorylation of Smad3 were not affected by Smad3 overexpression (Fig 2O–Q) could imply that the latter is not sufficient to activate RSmad-dependent TGFβ1 signaling, and hence a Smad3-dependent regulation of Bmal1 cannot be completely excluded.

## CDKO mice show an aggravation in UUO-induced fibrosis

The molecular clock impairment observed after kidney damage raises the question on the relevance of this alteration in the repair process. To answer it, we used different mouse models for molecular clock disruption: *Cry1, Cry2* single KO, *Cry1⁺/⁻Cry2⁺/⁻* (CDHet), *Cry1⁻/⁻Cry2⁻/⁻* (CDKO), *Clock^Δ19* mutant mice and a conditional mouse model with a global genetic deficiency for Bmal1 (Bmal1 KO). As expected, global Bmal1 KO mice showed alterations in their activity patterns under constant darkness (Fig S4A), and the expression of the different clock components was altered in kidneys from all mutant mice compared with their respective WT counterparts (Fig S4B–D). The analysis of kidney fibrosis in CDKO, CDHet, *Cry1,* and *Cry2* single KO mice 3 d after UUO (Fig 3A) revealed that the combined absence of the two *Cry* genes was associated with a more severe histological phenotype, observed by a significant increase of Kim1 expression and with a higher collagen deposition analyzed by Sirius red (Fig 3B and C) that correlated with a significant increase of soluble collagen (Fig 3D) and of its protein and mRNA levels (Fig 3E–G). Moreover, although the protein expression of fibronectin did not change significantly (Fig 3E and F), an increment in the protein levels of SMA and in mRNA levels of *Fn1, Acta2,* and other fibrosis-related genes was observed in CDKO compared with WT mice (Fig 3E–G). The assessment of kidney function in WT and CDKO mice revealed a slight increase in BUN and no significant changes in blood creatinine levels in CDKO 3 d after UUO (Fig 3H), as renal function is preserved by the non-ligated kidney. The analysis of changes in CDHet, *Cry1* KO, and *Cry2* KO mice revealed a significant increase in the protein levels of *Col1a1* in CDHet mice. However, no other significant differences were detected in these genotypes (Fig 3D–G).

The same molecular and histological analysis was performed in Bmal1 KO (Fig S5A–I) and *Clock^Δ19* (Fig S6A–H) mice subjected to UUO for 3 or 7 d. However, the results did not reveal significant differences regarding kidney fibrosis.

## *Cry1* and *Cry2* double–KO mice show an increased infiltration of neutrophils whereas the *Clock^Δ19* mice show increased number of macrophages after kidney damage

Fibrosis is the result of a maladaptive repair response to chronic inflammation. Thus, it is important to understand how modifications

---

**Figure 1. The molecular clock pathway is up-regulated in altered PT cells after kidney damage.**
**(A)** Schematic of experimental design of WT mice subjected to unilateral ureteral obstruction (UUO) for 3, 7, 15, and 25 d. **(A, B)** Heat map represents the mRNA expression of molecular clock genes and fibrotic markers in kidneys from mice described in (A) (n = 6 per condition), red rectangles denote significant differences compared with contralateral kidneys (Wilcoxon test). **(B, C)** Spearman r correlation of the data represented in (B), *P < 0.05. **(D)** Schematic of experimental design of the scRNA-seq data obtained from the dataset GSE182256. **(E)** UMAP dimension reduction of cells showing 22 distinct cell types identified by unsupervised clustering. GEC, glomerular endothelial cell; ALOH, ascending loop of Henle; DCT, distal convoluted tubule; CNT, connecting tubule; CD PC, collecting duct principal cell; A-IC, α-intercalated cell; B-IC, β-intercalated cell; prolif PT, proliferating PT. **(F)** Feature plot shows the expression of Arntl and Cry1 in the different cell types identified. **(G)** Heat map represents the up-regulated and down-regulated gene sets in immune cells from sham and UUO kidneys. **(H)** Violin plots represent the expression of molecular clock genes in immune cell–related clusters. **(I)** Bar plots represent the distribution of immune cells regarding the molecular clock gene expression. **(J)** UMAP plots showing the expression of molecular clock genes in PT cells from UUO kidneys. **(K)** Violin plots represent the expression of molecular clock genes in PT and PT Circadian cells, #P < 0.01, ###P < 0.001 compared with PT cells from control; *P < 0.05, **P < 0.01, ***P < 0.001 compared with PT cells from UUO (Wilcoxon test). **(L)** Density plot shows the expression and distribution of "KEGG-CIRCADIAN-RHYTHM" in PT cells from UUO kidneys in Ucell on UMAP plot. **(M)** Dot plot represents the expression of proximal tubular epithelial cells markers in PT circadian and PT cells from UUO. **(N)** Gene Ontology annotation of the top up-regulated and down-regulated pathways in PT circadian versus PT cells from UUO kidneys. The data were obtained from DAVID and represented as the −log(P-Value).

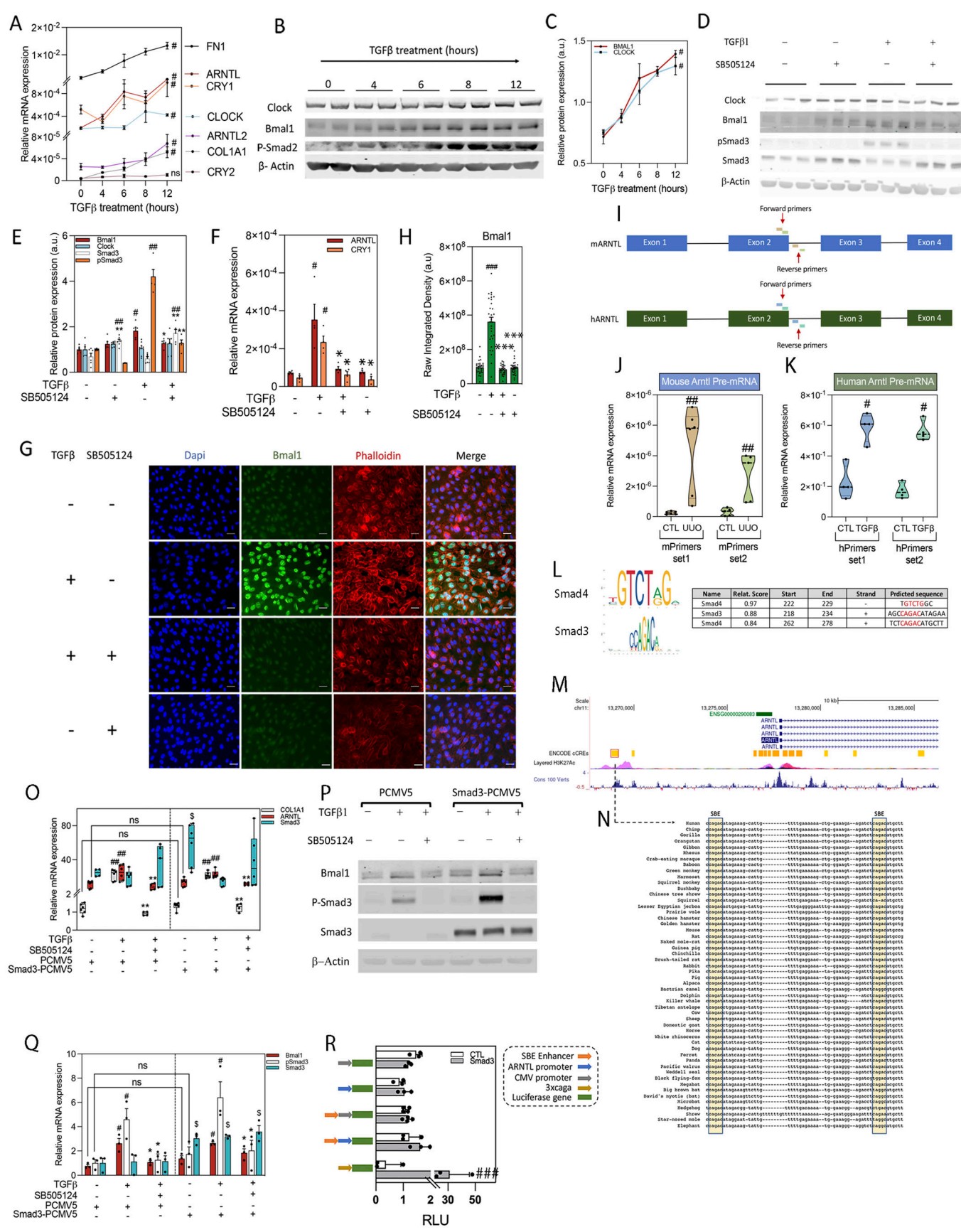

in the circadian clock may contribute to alter the response of the cellular inflammatory context that usually precedes and accompanies fibrosis. We attempted to clarify the role of the circadian clock in the inflammatory response that usually precedes and accompanies fibrosis. We first evaluated the abundance of innate immune cells in CDKO mice 3 d after UUO (Fig 4A). Flow cytometry analysis of CD11b⁺F4/80⁻ cells revealed a significant increase in the population SSC$^{int}$GR1$^{high}$ whereas no differences were observed in SSC$^{low}$GR1$^{low-neg}$ in CDKO compared with WT mice (Fig 4B and C), phenotypes that correspond to neutrophils and monocytes, respectively (21). These data correlate with the analysis of mRNA levels of Ly6c and Ly6g markers showing that only CDKO mice exhibited a significant increase in both control and obstructed kidneys, in contrast to Cry1 and Cry2 single KO or the CDHet (Fig 4E). The analysis of the CD11b⁺F4/80⁺ cell population did not reveal significant differences in the number of macrophage subpopulations positive for CD86 (M1) or for CD206 (M2) (Fig 4B and D). Moreover, the expression of several macrophage-related markers did not reflect significant differences among the genotypes analyzed (Fig 4F). Overall, data related to the CDKO phenotype support that (a) in the specific fibrosis model of UUO, the absence of Cry1/2 is a major determinant for phenotypic aggravation, whereas the latter is spared in mice genetically deficient for Bmal1 or Clock; (b) the simultaneous absence of Cry1 and Cry2 selectively affects the infiltration of neutrophils in the early inflammatory response accompanying UUO.

We next evaluated the inflammatory response in Clock$^{Δ19}$ mice 3 d after UUO (Fig 5A). We found that kidneys from Clock$^{Δ19}$ mice showed a higher number of the pro-inflammatory macrophages F4/80⁺CD86⁺ (Fig 5B and C). Similarly, the monocyte population F4/80⁻Ly6c⁺ was also increased in these animals compared with WT mice, although no differences were observed in the neutrophil subset

F4/80⁻Ly6c+Ly6g+ (Fig 5B and D). Consistently, immunohistochemical evaluation of F4/80 confirmed the increased presence of positive cells (Fig 5E and F). The mRNA expression of early inflammation markers and pro-inflammatory cytokines (Tnfa, IL-1b, IL-6, Ifng) was increased in renal tissue of Clock$^{Δ19}$ mice 3 d after UUO (Fig 5G). The CD86 and CD206 double-positive subset of macrophages—a transitional phenotype from a pro-inflammatory to anti-inflammatory state—was also increased in these mutant mice compared with WT (Fig 5B, C, and G). The number of CD86⁻CD206⁺ anti-inflammatory macrophages remained invariable between genotypes (Fig 5B and C). Similarly, Th2 anti-inflammatory cytokines (IL-4, IL-10) were undetectable (data not shown). In contrast, the analysis of the inflammatory response in Bmal1 KO mice 3 d after UUO (Fig S5A) did not reveal differences compared with WT mice (Fig S5J–L).

Studies in the late phase of inflammation were also performed in WT, Bmal1 KO, and Clock$^{Δ19}$ mice 7 d after UUO (Figs 5A and S5A) and in WT and Clock$^{Δ19}$ mice subjected to ADN for 25 d (Fig 5H). We observed that 25 d after the start with the ADN diet, the loss of Clock was associated with an increased presence of anti-inflammatory CD86⁻CD206⁺ and of CD86⁺ CD206⁺ macrophages. On the contrary, no significant differences were observed in the pro-inflammatory CD86⁺CD206⁻ macrophages, F4/80⁻Ly6c⁺ monocytes, and F4/80⁻Ly6c⁺Ly6g⁺ neutrophils (Fig 5I–K). In consistence, we found increased mRNA expression of anti-inflammatory markers (Arg1, Mrc1, and Il-17) in Clock$^{Δ19}$ mice 7 d after UUO (Fig 5L), whereas no significant differences were observed in F4/80 immunohistochemistry (IHC) (Fig 5E and F). In contrast, but consistent with the data obtained in the early phase of inflammation, Bmal1 KO did not show significant differences 7 d after UUO (Fig S5J, K, and M). These data further support an involvement of Clock, rather than Bmal1, in the early and late phases of the macrophage inflammatory response inherent to the UUO model.

---

**Figure 2. The expression of molecular clock components is up-regulated by TGF-β in proximal tubular epithelial cells.**

**(A, B, C)** Evaluation of the expression of molecular clock genes and fibrotic markers in synchronized human proximal TEC (HPTEC) treated with TGF-β for 0, 4, 6, 8, and 12 h. **(A)** Relative mRNA levels of molecular clock genes, Col1a1 and Fn1. **(B)** Immunoblots showing the Clock, Bmal, and pSmad2 expressions. β-actin was used for normalization. **(C)** Relative protein expression of Bmal1 and Clock obtained by densitometry of images from C and normalized with β-actin. **(D, E, F, G, H)** Evaluation of the expression of Bmal1, Cry1, and Clock in synchronized HPTEC treated with TGFβ for 24 h in the presence or absence of the selective inhibitor SB505124. **(D)** Immunoblot depicting the expression of Clock, Bmal1, pSmad3, and Smad3. β-actin was used for normalization. **(E)** Bar plot represents the relative protein expression of Bmal1, Clock, Smad3, and pSmad3 obtained by densitometry of images from D and normalized with β-actin. **(F)** Bar plot represents the levels of relative mRNA of Bmal1 and Cry1. **(G)** Immunofluorescence microscopy of Bmal1; Dapi was used for the staining of the nucleus and phalloidin for the actin cytoskeleton. Scale bar: 50 μm. **(H)** Quantification of the fluorescence of Bmal1 calculated from 10 different fields per sample. The experiment was repeated three times, and the data are represented with bar plots as the mean of the integrated density of all fields analyzed (a total of 30 fields per condition) ± s.e.m. **(I)** Schematic depicting the amplification regions of the primers designed for the analysis of the expression of the mouse (blue) and human (green) pre-mRNA of the gene Arntl. **(J)** Relative pre-mRNA levels of Arntl in contralateral and obstructed kidneys from mice 25 d after unilateral ureteral obstruction (UUO). **(K)** Relative pre-mRNA levels of Arntl in synchronized human primary proximal tubular epithelial cells treated with TGF-β for 24 h. **(L)** Smad3 and Smad4 (positive and negative strands) sites predicted in the distal enhancer by JASPAR2018 with a relative score above 80%. **(M)** Representation denoting the regulatory regions located upstream of the ARNTL gene. Data obtained from the UCSC genome browser (19). The image shows the DNA sequence upstream of the human ARNTL gene and indicates distal and proximal enhancers, H3K27Ac marks and conservation among vertebrates. **(N)** Representation denoting the conservation among vertebrates of the R-Smads potential binding sites found in a distal enhancer close to the ARNTL gene. Data were obtained with the UCSC genome browser (19). **(O)** Relative mRNA levels of Col1a1, Arntl, and Smad3 in synchronized HPTEC transfected with the overexpression vector for Smad3 (Smad3–PCMV5) or the control vector (PCMV5) and treated with TGF-β for 24 h in the presence or absence of the selective inhibitor SB505124. **(O, P)** Immunoblots showing the Bmal1, pSmad3, and Smad3 expressions under the same conditions as (O). β-actin was used for normalization. **(Q)** Relative protein expression of Bmal1, pSmad3, and Smad3 obtained by densitometry of images from P and normalized with β-actin. **(R)** Bioluminescence assays in HPTEC transfected with the vector reporter constructions Pgl3p, hARNTLp, SmEnh-Pgl3p, SmEnh-hARNTLp, 3xCAGA, and with the overexpression vector for Smad3 treated with TGFβ for 24 h in the presence or absence of the selective inhibitor SB505124. **(A, C, E, F, J, K, O, Q, R)** Data information: data represent the mean ± s.e.m of (A, F, K) n = 4, (C, Q, R) n = 3, or (E, J, O) n = 6 independent experiments; each performed in triplicates. **(E, F, G, H, I, J, K, L, M, N, O, P, Q, R)** #P < 0.05, ##P < 0.01, ###P < 0.001 compared with control without treatments, *P < 0.05, **P < 0.01, ***P < 0.001 compared with cells treated with TGFβ only, $P < 0.05 compared with their corresponding control without Smad3 overexpression (Mann–Whitney). **(A, C)** #P < 0.05 significant throughout time (one-way ANOVA). Source data are available for this figure.

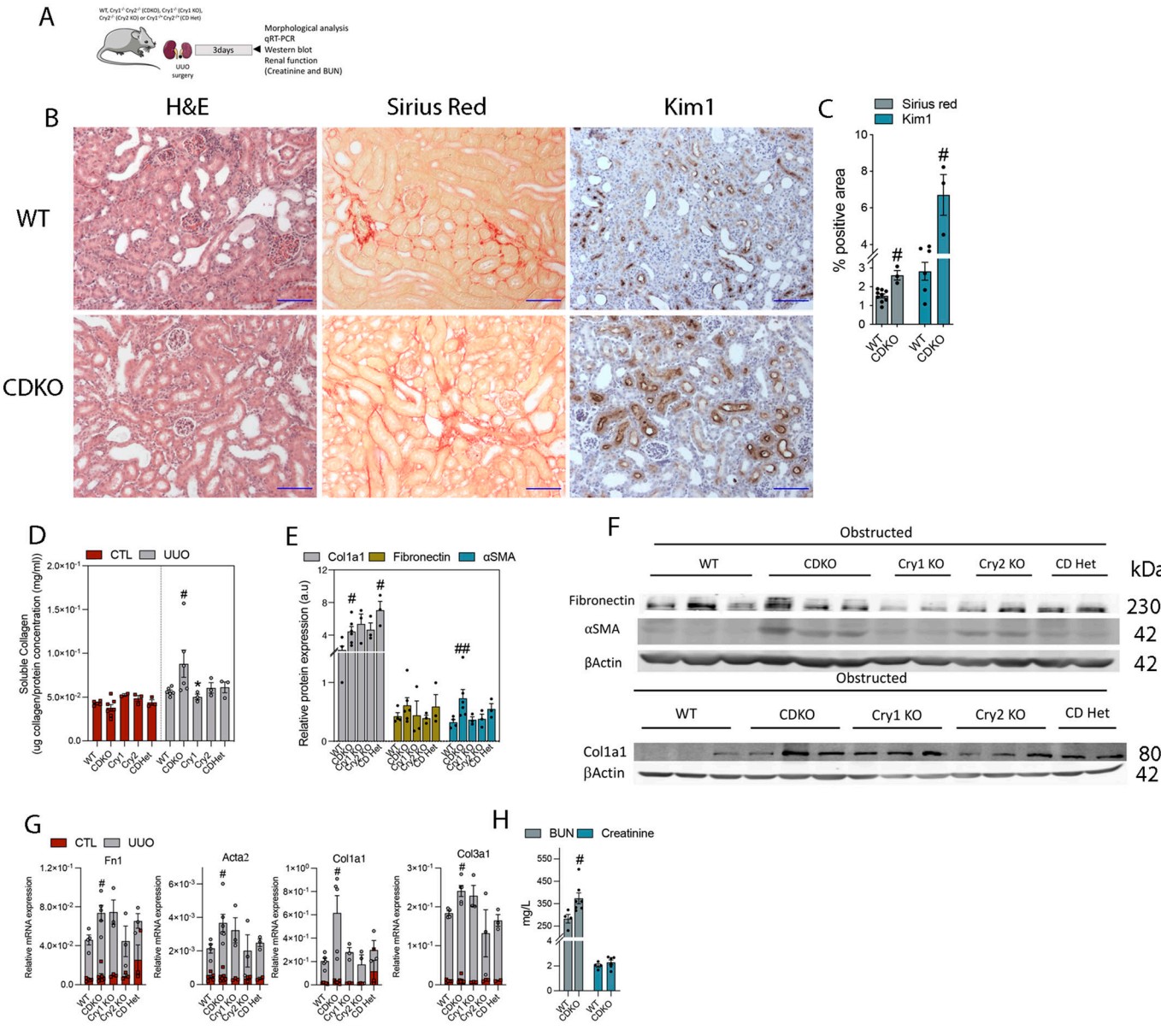

**Figure 3. Fibrosis is exacerbated by Cry 1/2 genetic deficiency.**
**(A)** Schematic of experimental design of WT and Cry1/Cry2 KO mice subjected to unilateral ureteral obstruction (UUO) for 3 d. **(B)** Representative microphotographs of H&E, Sirius red, and KIM1 immunohistochemistry stains from obstructed kidneys of WT and CDKO. Scale bar: 150 μm. **(C)** Quantification of Sirius red and KIM1 immunohistochemistry in kidney sections of WT and CDKO mice (Sirius red number of mice: WT [8], CDKO [3]; KIM1 number of mice: WT [6], CDKO [3]). **(D)** Bar plot represents the quantification of the total soluble collagen in frozen kidneys from WT, CDKO, Cry1 KO, Cry2 KO, and CDHet mice. **(E, F)** Bar plot represents the relative protein expression of Col1a1, fibronectin, and SMA of images in (F). **(F)** Immunoblot depicting the expression of fibronectin, αSMA, and Col1a1 in kidneys from WT, CDKO, Cry1 KO, Cry2 KO, and CDHet mice subjected to UUO for 3 d, β-actin levels were used for normalization. **(G)** Relative mRNA expression of fibrosis-related genes in WT, CDKO, Cry1 KO, Cry2 KO, and CDHet mice subjected to UUO. **(H)** BUN and plasma creatinine levels of WT and CDKO mice 3 d after UUO. Number of mice: Cry-deficient mice: WT (n = 6), CDKO (n = 6), Cry1 KO (n = 3), Cry2 KO (n = 3), CDHet mice (n = 3); #$P < 0.05$, ##$P < 0.01$ compared with the WT (Mann–Whitney).
Source data are available for this figure.

## The expression of metabolism-related genes is differentially affected by the selective deletion of molecular clock components in the UUO model of kidney fibrosis

In the past years, the role of metabolic failure of TECs in the genesis and progression of kidney disease has been clearly established

(6, 7). As we previously observed, the enrichment of the molecular clock pathways in PT cells after kidney damage is correlated with a metabolic derangement (Fig 1N). Because of the connection between circadian rhythm and metabolism, we sought to determine the effect of the different clock mutant mice on the expression of key metabolic genes in the UUO model of kidney fibrosis.

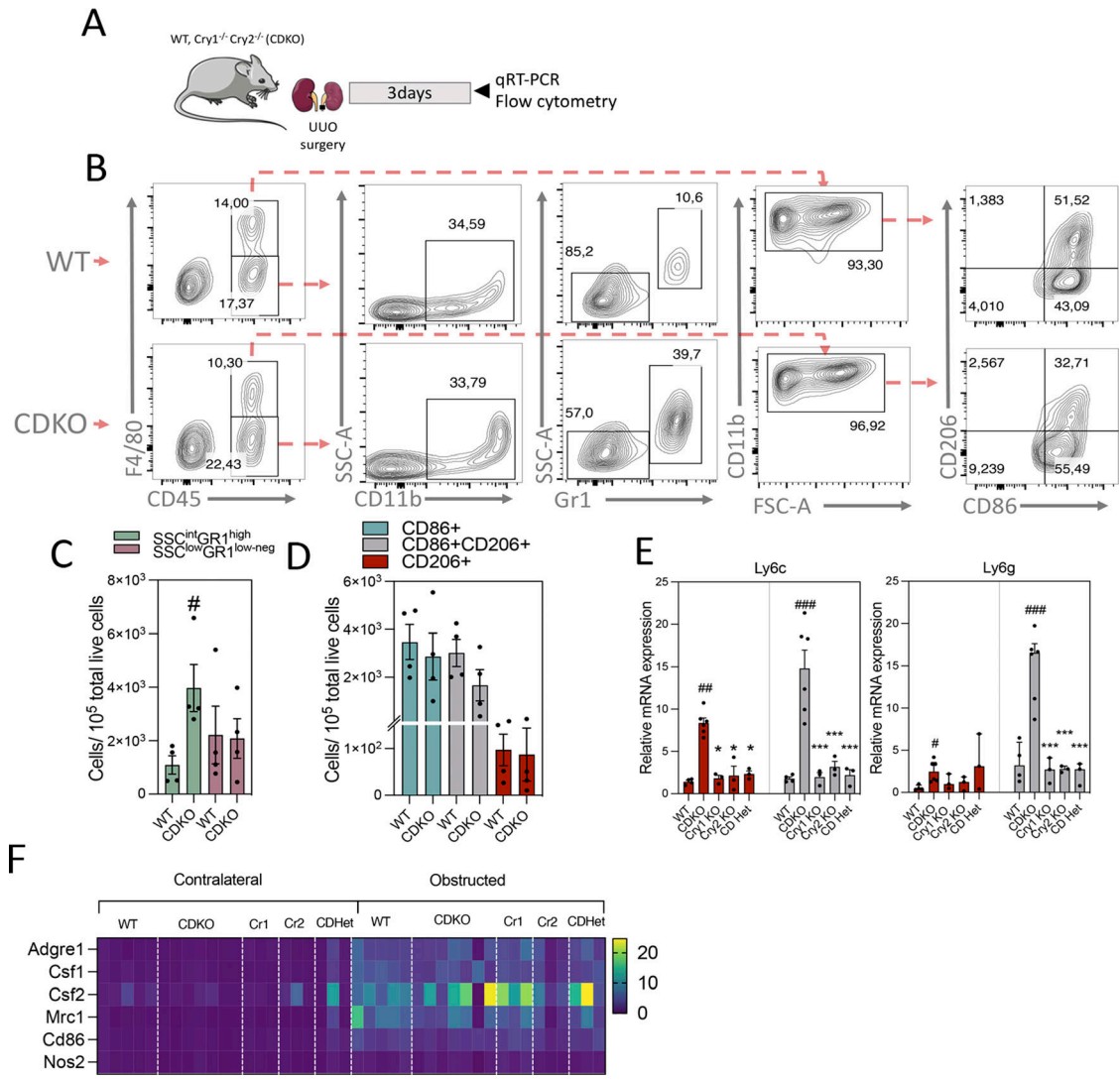

**Figure 4. The inflammatory response of CDKO mice is associated with an increased presence of neutrophils.**
**(A)** Schematic of experimental design of WT Cry1/Cry2 KO mice subjected to unilateral ureteral obstruction (UUO) for 3 d. **(B)** Flow cytometry dot plots denoting the percentage of myeloid cell populations in kidneys from WT (n = 4) and CDKO (n = 4) mice. **(B, C)** Bar plots represent the FACS quantification of the number of neutrophils (SSC$^{int}$GR1$^{high}$) and monocytes (SSC$^{low}$GR1$^{low-neg}$) from the analysis shown in (B). **(B, D)** Bar plots represent the FACS quantification of the number of different macrophage subpopulations based on the expression of CD86 and CD206 from the analysis shown in (B). **(E)** Bar plots represent the relative mRNA expression of Ly6c and Ly6g in kidneys from WT, CDKO, Cry1 KO, Cry2 KO, and CDHet mice subjected to UUO for 3 d. **(F)** Heat map of normalized expressions of macrophage-related markers in kidneys from WT, CDKO, Cry1 KO, Cry2 KO, and CDHet mice subjected to UUO for 3 d. Number of mice: WT (n = 5), CDKO (n = 6), Cry1 (n = 3), Cry2 (n = 3), and CDHet (n = 3) mice. $^{#}P < 0.05$, $^{##}P < 0.01$, $^{###}P < 0.001$ compared with the WT. $^{***}P < 0.001$ compared with the CDKO (Mann–Whitney).

Specifically, we analyzed genes involved in five major metabolic pathways, including those related to mitochondrial function and bioenergetics. We found that *Cry1/Cry2*-deficient mice showed a significant reduction in transcripts of all these five metabolic routes both in non-obstructed and obstructed kidneys (Fig 6A). Consistent with this transcriptomic profiling, Tfam protein levels were diminished in contralateral kidneys of the CDKO, *Cry1*KO, and CD heterozygous mice and in CDKO and CD heterozygous after UUO, compared with WT, whereas no significant differences were observed in Cpt1a protein expression (Fig 6B–D). By contrast, the number of genes with significant reduction in their expression was much lower in the case of the Bmal1 KO mice, mostly affecting FAO-related genes

(Fig S7A–C). Similarly, the pattern of reduction in the expression of metabolism-related genes was not sustained in the *Clock*$^{Δ19}$ mice (Fig S7D–F), thus suggesting that the deletion of *Cry1* and *Cry2* proteins is specifically associated with a metabolic derangement that can contribute to kidney damage.

## Discussion

Kidney function is intrinsically linked to the circadian rhythm, exemplified by oscillations in hormone secretion and blood pressure (22, 23, 24, 25). In consequence, kidney dysfunction and in particular

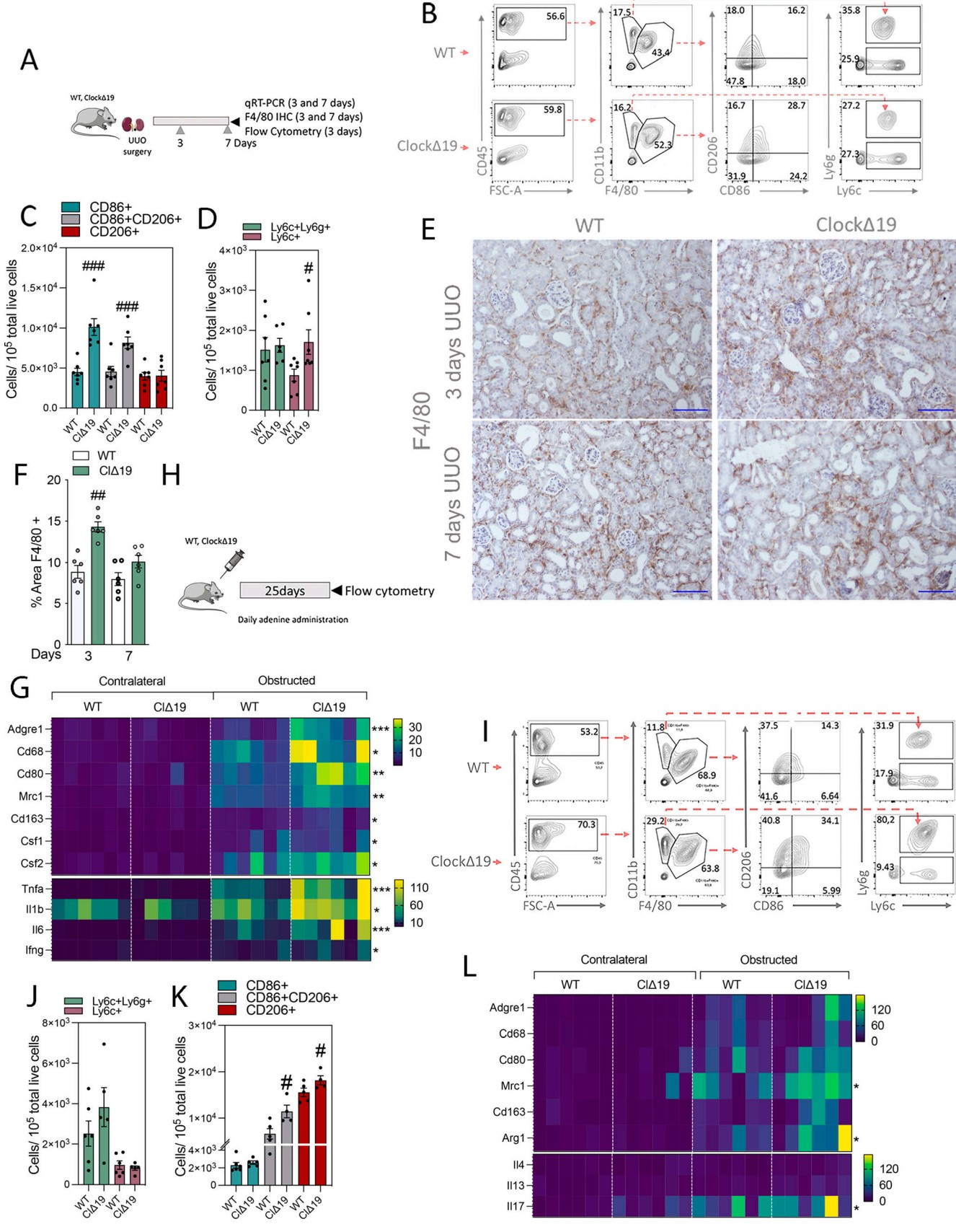

CKD have been directly associated with perturbations of the circadian rhythm (12, 26, 27, 28, 29). Nevertheless, the mechanisms for CKD progression and its major underlying condition, fibrosis, in relation to specific disturbances of the molecular clock are incompletely understood. Our results demonstrate a disruption of the peripheral molecular clock in the kidneys as a consequence of renal damage. Interestingly, the KEGG molecular clock pathway is enriched in those populations of PT cells that exhibit alterations in energy metabolism and with a defined cell damage molecular signature. Moreover, we found that TGFβ1 increased the expression of core clock genes in HPTEC in a time-dependent fashion. The study of several molecular clock mutant mice models did not demonstrate a preponderant role for Bmal1 in inflammation or fibrosis. By contrast, the disruption of the molecular clock components *Clock* and *Cry* was important for the inflammatory response related to macrophages and neutrophils, respectively. Moreover, the combined absence of both Cry proteins resulted in a specific modification of the metabolism-associated transcriptomic profile and a worsening of fibrosis.

Our results are consistent with other reports that found a perturbed pattern in the rhythmic oscillations of circadian genes in injured kidneys (11, 30, 31, 32). In addition, previous studies have revealed that the TGFβ pathway is able to modify the oscillatory patterns of core clock genes from different tissues (33, 34, 35, 36), and other authors have reported a reduction in the expression of Bmal1 in heart tissue of Smad3 knockout mice (37) and a role for Smad4 in pancreatic cancer cells (38) increased transcriptional activity of several potential cis-regulatory elements after Smad3 overexpression. However, the specific mechanism has not yet been clarified. Although we did not find them, we cannot exclude the participation of other as yet, unidentified enhancers. Hence, further studies are needed to decipher the molecular mechanisms responsible for molecular clock up-regulation in kidney fibrosis.

A persistent inflammatory response leads to the perpetuation of repair processes that involve a gradual instauration of fibrosis (39). To our knowledge, this is one of the first studies documenting the relevance of these core clock genes in kidney inflammation. In other pathological contexts, Bmal1 seems to play a role in the metabolic switch of macrophage populations (40, 41, 42). However, Bmal1 was not critical for the inflammatory instauration associated to lung and liver disease (43, 44), and in this study, the expression of cytokines and inflammatory markers was not affected in the global Bmal KO after UUO (Fig S5). As we analyzed time points compatible with inflammatory (3 and 7 d) or pre-fibrotic changes (7 d), we

cannot exclude that in a later phase, likely associated with a full fibrotic kidney, the absence of Bmal could play a role. In this regard, in a postnatal inducible model of Bmal1 suppression, reduced levels of tubulointerstitial fibrosis were found after 7 d of UUO, an effect mediated by Gli2 (36), suggesting the timing of Bmal1 deletion might play a key role. The *Clock*$^{Δ19}$ mutation has been associated with a pro-inflammatory phenotype in non-alcoholic liver disease (45). Clock has been identified as a protein implicated in the NF-κB–dependent inflammatory response by its direct interaction with the NF-κB subunit, p65, which competes with Bmal1 (46). These authors observed that *Clock* KO mice showed a decrease in the accumulation of NF-κB in hepatocyte nuclei, a phenomenon absent in the *Clock*$^{Δ19}$ genotype, suggesting a Clock-mediated circadian-independent regulatory mechanism of inflammation. Although these studies are consistent with our observations, the absence of significant changes in the fibrotic phenotype observed by us and the variability in the results reported by other authors (11, 43, 45, 47, 48) could imply that Clock-mediated inflammatory response is unable by itself to promote a drastic fibrotic phenotype. Neutrophils are subjected to circadian regulation as illustrated by diurnal variations in their levels (49). However, the relevance of the different core clock components in this regulation is poorly understood. To our knowledge, a specific role for Cry proteins in the regulation of neutrophil-mediated inflammatory responses has not been previously reported. Of interest, increased neutrophilia beyond the acute phase of inflammation has been linked to enhanced renal fibrosis (50, 51). In keeping, in consistence with our observations, a massive infiltration of leukocytes into the lungs and kidneys has been previously reported in CDKO mice as a consequence of an autoimmune response through the modulation of the BCR-signaling pathway (52). Whether this dysfunctional B-cell response contributes to the observed aggravation of kidney fibrosis remains to be investigated. Moreover, Cry proteins have also been associated with the inhibition of the NF-κB–mediated inflammatory response (53).

In the literature, variable results have been reported regarding the role of Bmal1 and Clock in the instauration of fibrosis. Although a profibrotic role of these proteins was observed in some studies in the context of kidney, lung, and heart disease (30, 34, 35, 36), Bmal1 and Clock appear to exert a protective role in other studies related to kidney, lung, and liver fibrosis (11, 43, 45, 47, 48, 54). Of interest, the deletion of Clock was associated with aggravation of fibrosis in the UUO model, and in this same study TGFβ appeared to be regulated by Clock–Bmal1 heterodimers (31). Our study now incorporates

**Figure 5. ClockΔ19 mice exhibit an increase in the number of macrophage and monocyte populations in the early and late phases of inflammation.**
**(A)** Schematic of experimental design of WT and Clock$^{Δ19}$ mice subjected to unilateral ureteral obstruction (UUO) for 3 d. **(B)** Flow cytometry dot plots denoting the percentage of myeloid cell populations in kidneys from WT (n = 6) and Clock$^{Δ19}$ (n = 6) mice. **(B, C)** Bar plots represent the FACS quantification of the number of different macrophage subpopulations based on the expression of CD86 and CD206 from the analysis shown in (B). **(B, D)** Bar plots represent the FACS quantification of the number of neutrophils (Ly6c$^+$Ly6g$^+$) and monocytes (Ly6c$^+$Ly6g$^-$) from the analysis shown in (B). **(E)** Representative pictures of F4/80 immunohistochemistry stain of contralateral and obstructed kidneys of WT and Clock$^{Δ19}$ mice 3 and 7 d after UUO. Scale bar: 150 µm. **(E, F)** Quantification of F4/80 immunohistochemistry in obstructed kidney sections of mice denoted in (E). **(G)** Heat map of normalized expressions of inflammatory-related genes in kidneys from WT and Clock$^{Δ19}$ mice 3 d after UUO. **(H)** Schematic of experimental design of WT and Clock$^{Δ19}$ mice subjected to adenine-induced nephropathy for 25 d. **(I)** Flow cytometry dot plots denoting the percentage of myeloid cell populations in kidneys from WT (n = 5) and Clock$^{Δ19}$ (n = 6) mice. **(I, J)** Bar plots represent the FACS quantification of the number of neutrophils (Ly6c$^+$Ly6g$^+$) and monocytes (Ly6c$^+$Ly6g$^-$) from the analysis shown in (I). **(I, K)** Bar plots represent the FACS quantification of the number of different macrophage subpopulations based on the expression of CD86 and CD206 from the analysis shown in (I). **(L)** Heat map of normalized expressions of inflammatory-related genes in kidneys from WT and Clock$^{Δ19}$ mice 7 d after UUO. $^{#}$/*P < 0.05, $^{##}$/**P < 0.01 $^{###}$/***P < 0.001 compared with WT (Mann–Whitney).

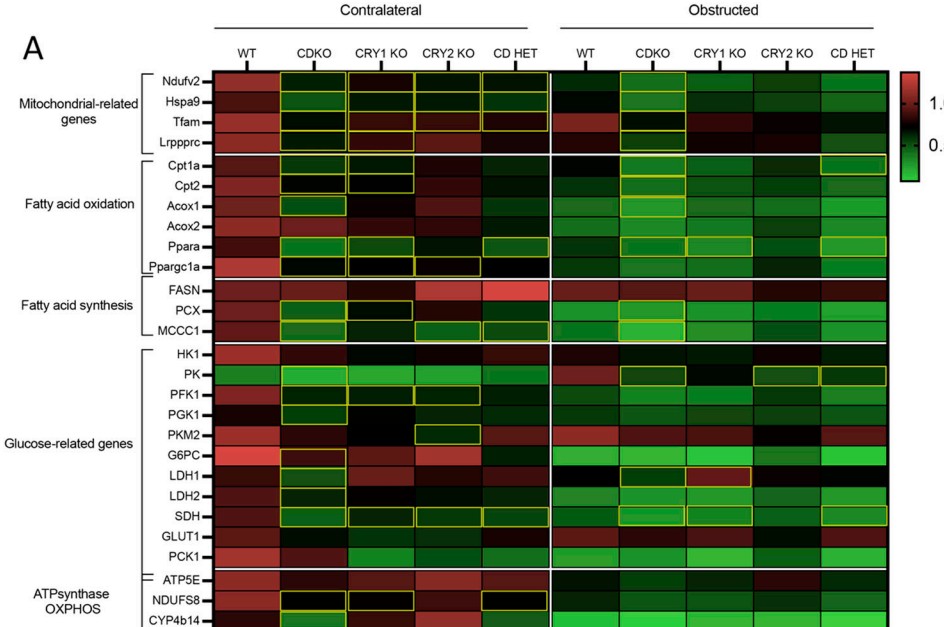

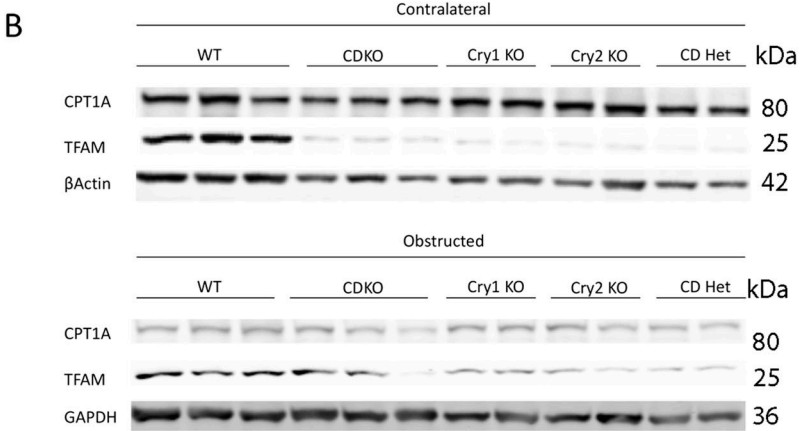

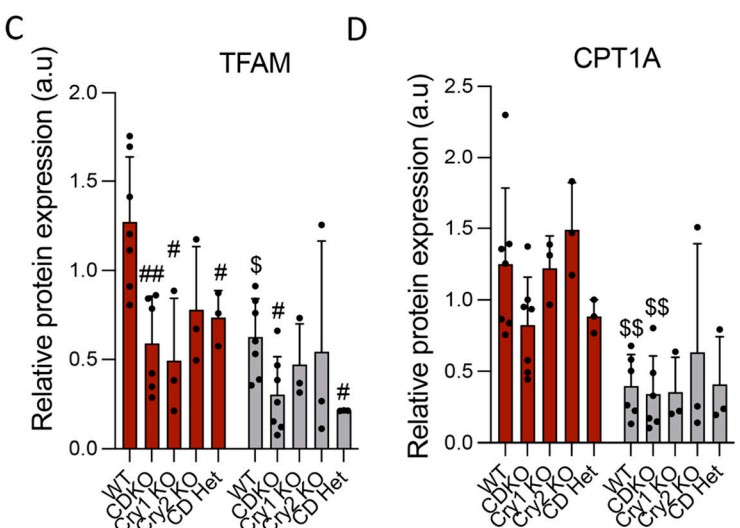

**Figure 6. The expression profile of metabolism and mitochondrial-related genes is significantly altered in CDKO mice.**
**(A)** Heat map of normalized expressions of metabolism and mitochondrial-related genes in CDKO mice, in kidneys 3 d after unilateral ureteral obstruction. **(B)** Immunoblot depicting the expression of Cpt1a and Tfam proteins in kidneys from mice with the different Cry1- and Cry2-defective genotypes, 3 d after unilateral ureteral obstruction; $\beta$-actin and GAPDH levels were used for normalization. **(B, C, D)** Relative protein expression of (C) TFAM and (D) CPT1A from the immunoblots shown in (B). Number of mice: WT (n = 5), CDKO (n = 6), Cry1 KO (n = 3), Cry2 KO (n = 3). **(A)** Yellow rectangles denote significant differences compared with kidneys from WT mice (Mann–Whitney). **(C, D)** $^{\$}P < 0.05$, $^{\$\$}P < 0.01$ compared with their respective contralateral kidney; $^{\#}P < 0.05$, $^{\#\#}P < 0.01$ compared with the WT (Mann–Whitney).
Source data are available for this figure.

another component of the molecular clock, Cry, as an important regulator of kidney damage. Our observations in the CDKO mice are concordant with studies in other fibrotic contexts. Hence, mice with a deletion of either *Cry1* or *Cry2* developed a more severe phenotype in a barium chloride–induced model of muscle fibrosis (55) and CDKO mice have shown altered expression patterns of aldosterone, a vital extrarenal hormonal regulator of kidney-mediated hydro-electrolytic homeostasis (56).

Mitochondrial bioenergetic dysfunction is involved in the development of several diseases (57, 58, 59, 60). A recent remarkable study performed in a mouse model with inducible deletion of Bmal1 in the renal tubule demonstrated a link between the disruption of circadian rhythmicity and profound metabolic alterations, including mitochondrial function, FAO, and carnitine transport (61). Although this work did not evaluate the fibrotic outcome, the metabolic derangement is fully consistent with our previous work showing protection from fibrosis by enhanced FAO (7). Noteworthy, the present study revealed that the levels of TFAM, a key activator of mitochondrial transcription, were dramatically reduced in CDKO mice. Tubule-specific deletion of TFAM has been related to an increase in renal fibrosis (62), thus supporting the profibrotic phenotype found in these mice. Although these results suggest a link between Cry function and kidney metabolism, the role of Cry proteins in the metabolic derangement associated with fibrosis should be further explored. Taken together, our results support there is a reciprocal influence between kidney injury and several components of the molecular clock. Delving into this crosstalk may provide new avenues for the identification of CKD progression biomarkers and therapeutic targets.

# Materials and Methods

## Genetically modified mouse models in the circadian clock (clock-mutant mice)

*Clock*$^{\Delta19}$ mice, *Per2*::Luc mice (kindly provided by Dr. Takahashi's laboratory, TX, USA), Bmal1$^{lox/lox}$/Pax8-rtTA/LC1 (Bmal1 cKO), Bmal1$^{lox/lox}$/CRE-ER (Bmal1 KO), and *Cry1*/*Cry2*-mutant mice (double-KO mice [CDKO], *Cry1* KO, *Cry2* KO, and double-heterozygous [CDHet]) were bred on the genetic background of the C57BL/6J mouse and were generated and characterized as previously described (14, 15, 16, 17). *Clock*$^{\Delta19}$ mice were crossed with *Per2*::Luc mice, using the *Per2*::Luc strain as control (WT), Bmal1$^{lox/lox}$ mice (WT) were used as control for the Bmal1 cKO and the Bmal1 KO and male C57BL/6J mice (WT) purchased from the Jackson Laboratory were used as controls for the CDKO mice. The genotype of the animals was confirmed by PCR by using the primers listed in Table S2. The animals were maintained ad libitum on regular chow diet, under constant temperature and humidity in a 12-h light/dark cycle. Doxycycline and tamoxifen were used for the induction of Cre expression in 8 wk-old Bmal1 cKO and Bmal1 KO mice, respectively. Mice were treated daily with 2 mg/ml of doxycycline in drinking water (17) or 3.7 mg of tamoxifen administered by oral gavage for 2 wk or 5 d, respectively. Subsequent experiments were performed 4 or 2 wk after the end of the treatment, respectively, to avoid potential side effects

related to their administration. Mouse activity was tested in the Bmal1 KO and their WT mice after Cre expression induction with tamoxifen using the Mouse Home Cage Running Wheel (the Columbus Instruments). To make sure that tamoxifen effects on inflammation were not interfering with the experimental conditions used, we made sure that the expression of panoply of inflammatory markers were not affected by tamoxifen by itself (data not shown).

## Mouse models of CKD

UUO procedure was performed as described previously (18). Briefly, 12–14 wk-old mice were anesthetized with 2% isoflurane. The hair in the abdominal area was shaved. An incision was made in the abdominal wall to expose the left kidney, the ureter was ligated twice and severed between the two ligatures, and the kidney was returned gently to its place. Finally, the abdominal incision was closed with sutures and buprenorphine was used as an analgesic. FAN was carried out as previously described (20). Mice were given 250 mg/kg of folic acid intraperitoneally (i.p) (Sigma-Aldrich) resuspended in 0.3 M sodium bicarbonate (vehicle) and control mice were given 0.1 ml of the vehicle. For ADN-induced renal failure, mice were given 50 mg/kg of ADN in 0.5% carboxymethyl cellulose (Wako Pure Chemical Industries Ltd.) daily by oral gavage as described previously (21). Control mice were given 0.1 ml of 0.5% carboxymethyl cellulose (vehicle). Mice were euthanized 3, 7, 15, or 25 d after UUO; 7 and 15 d after FAN or 25 d after ADN. The three procedures and the euthanization of the mice were performed at ZT9 (where ZT0 means light on and ZT12 means light off). Blood samples were collected right after the euthanization by cardiac puncture, and kidneys were collected after perfusion with PBS. Number of mice is detailed in Table S3.

## Single-cell RNA-seq analysis

The mouse kidney single-cell data from the publicly available dataset GSE182256 and GSE212273 were aligned using CellRanger v.7.0.1 (10X Genomics) using mouse (GRCm38) genome reference. The filtered matrix obtained by CellRanger was used to make Seurat objects for each sample with genes expressed in more than three cells and cells with at least 300 genes by using Seurat (v.4.0.3) (63). The Seurat objects were merged and used for downstream analysis. The low-quality cells were filtered (cells with mitochondrial counts >50%, n_Feature_RNA < 200 or n_Feature_RNA > 3,000 t). The top variable genes were identified using the "vst" method (64). Next, the data were normalized and scaled in terms of n_Count_RNA. Harmony (v.1.0) (65) was used to remove batch effect. The results of harmony were projected to UMAP for clusterization. "FeaturePlot" function of the Seurat software was used for feature visualization on a dimensional reduction plot. "Vlnplot" and "DotPlot" were used for gene expression visualization. The Wilcoxon test was used to calculate the statistical differences in gene expression. To identify DEGs in each cell types, the "FindAllMarkers" function of the Seurat with the following parameters was used: logfc.threshold, 0.25, adjusted *P*-value < 0.05. DAVID was used for functional enrichment analysis in PT cells. The "irGSEA" package (v1.1.2) (https://github.com/chuiqin/irGSEA/) in R

software was used to assess the enriched KEGG gene sets. The Wilcoxon test was used to calculate the differentially expressed gene sets. "Scatterplot" and "irGSEA.heatmap.plot" were used for visualization.

## Cell culture

Human primary renal proximal tubule epithelial cells (HprimPTEC) and the immortalized human renal proximal tubule epithelial cells RPTEC/TERT1 (HPTEC) were obtained from the American Type Culture Collection (#PCS-400-010 and #CRL-4031; ATCC, respectively). These cells were cultured in renal epithelial cell basal medium (#PCS-400-030; AT ATCC) supplemented with the renal epithelial cell growth kit (#PCS-400-040; ATCC) that contains the following components: triiodothyronine (10 nM), rhEGF (10 ng/ml), hydrocortisone hemisuccinate (100 ng/ml), rh insulin (5 $\mu$g/ml), epinephrine (1 $\mu$M), L-alanyl-L-glutamine (2.4 mM), transferrin (5 $\mu$g/ml), 0.5% (v/v) FBS at 37°C and 5% $CO_2$.

Mouse primary kidney cells were isolated from C57BL6/J WT mice as follows: mice were euthanized with $CO_2$ overdose and kidneys were collected after perfusion with cold PBS. The capsule was removed and kidneys were minced and digested in PBS containing 2 mg/ml of collagenase from Clostridium histolyticum (Sigma-Aldrich) for 20 min at 37°C with gentle stirring, after which supernatants were sieved through a 70-$\mu$m nylon mesh. The cells were incubated with sterile red blood cell lysis buffer (BioLegend) and seeded in 10 cm culture dishes. Cells were cultured in RPMI 1640 (Corning) supplemented with 10% FBS, 20 ng/ml EGF (Sigma-Aldrich), 50 units/ml penicillin, and 50 $\mu$g/ml streptomycin (Gibco) at 37°C and 5% $CO_2$. In vitro experiments were performed as described below.

## In vitro experiments

HPTEC, HPrimPTEC, and mouse primary kidney cells were plated on six- well plates and were left on 2%, 0.5%, or 10% FBS fresh media, respectively, to reach 80% confluency. Then, cells were synchronized with 100 nM dexamethasone for 2 h before being washed twice with 1x PBS and placed in 0.5% FBS fresh media for 24 h. Cells were treated with human recombinant 10 ng/ml TGFß1 (R&D Systems), resuspended in 4 mM HCl 1 mg/ml BSA (vehicle) and/or 2.5 $\mu$M SB505124 (Sigma-Aldrich), resuspended in DMSO (vehicle) for 0 to 24 h in 0.5% FBS media before analyses. Control cells without TGFß1 and SB505124 treatment were treated with the same volume of the vehicles used for the resuspension of both compounds. HPTEC were transfected using Lipofectamine 2000 (Invitrogen) with the plasmid constructions described below. 3xCAGA-Luc vector was used as a positive control, Smad3–pCMV5 vector was used for Smad3 overexpression, pCMV5 vector was used as control (these vectors were kindly provided by Dr. Fernando Rodriguez-Pascual, Centro de Biología Molecular Severo Ochoa). pRL–CMV vector (Promega Corporation) containing WT renilla luciferase was used for normalization in reporter assays at a 1:30 ratio. Cells were incubated with lipofectamine and the corresponding plasmid for 6 h at 37°C. Subsequently, fresh media containing 0.5% FBS was added to the culture and cells were maintained at 37°C for 24 h until reporter analysis or were then treated with human recombinant 10 ng/ml TGFß1 and/or 2.5 $\mu$M SB505124 for an additional 24 h for subsequent mRNA and protein expression analysis.

## Assay for measuring cytotoxicity

The MTT assay (M-2128; Sigma-Aldrich) was used to measure cell viability. In brief, target cells were seeded in 96-well microplates to reach 80% confluency. Different dilutions, from 0 to 3.5 $\mu$M, of SB505124 were added to the target cells. The cells without the compound were treated with its corresponding vehicle. After 24 h of incubation at 37°C, MTT at a concentration of 5 mg/ml in PBS was added and further incubated for 4 h at 37°C. After aspirating the supernatant from the wells, isopropanol with 0.04 N HCl was added to all wells. After dissolving the dark blue formazan crystals, the plates were read on a GloMax-Multi Detection system (Promega), using a test wavelength of 570 nm and a reference wavelength of 630 nm. Percentage of viable cells was determined as per manufacturer's instructions. Results are presented as mean + SEM of sextuplicate determinations.

## Immunofluorescence

HPTEC were fixed in 4% neutral-buffered formalin for 10 min and permeabilized with 0.1% Triton X-100 in PBS for 10 min at RT. Next, they were blocked with 1% BSA in PBS for 30 min at RT and incubated overnight with the Bmal1 primary antibody listed in Table S4. Then, the samples were incubated for 1 h for staining with the fluorochrome-conjugated secondary antibody anti-mouse Alexa488 (Thermo Fisher Scientific). Actin cytoskeletons and nuclei were stained with phalloidin TRITC and DAPI for 40 and 5 min at RT, respectively. The coverslips were mounted on slides using MOWIOL (Calbiochem). Cell fluorescence was visualized by a CoolSNAP Fx Monochrome confocal microscope with a 40x/1.3 oil Plan-Neofluar M27 objective (Zeiss).

## Plasmid construction

In silico analysis was performed on human *ARNTL* regulatory region using the R package for JASPAR2018 (20). The distal enhancer E1522401 (SmEnh) of the human *ARNTL* gene (located from −9,325 to −8,376; where +1 bp is the putative transcription start site) was cloned into the Pgl3-promoter or Pgl3-basic vector (Promega Corporation). The human *Arntl* promoter region (h*Arntl*p; from −1,220 to +8) was used as the promoter sequence for the Pgl3-basic vector by cloning it downstream of the SmEnh sequence. For that purpose, human genomic DNA was obtained from HPTEC using the PureLink kit (Invitrogen) according to manufacturers' instructions. The inserts were amplified by PCR using the primers listed in Table S5 and the AccuStart kit (QuantaBio, VWR International) following the manufacturers' instruction. The PCR products were tested by 1% agarose gel electrophoresis (CONDA) and purified with QIAquick Gel Extraction Kit (QIAGEN). Once purified, the inserts were first cloned into PCR2.1 vector using the TA-Cloning kit (Invitrogen) and subsequently subcloned into the PGL3-basic or PGL3-promoter vectors. The construct was verified by Sanger sequencing and diagnostic enzymatic digestions with restriction enzymes.

## Reporter assays

HPTEC were lysed with passive lysis buffer (Promega) 24 h after transfection, and firefly and renilla luciferase activities were determined using a dual-luciferase reporter system (Promega Corporation) and measured using the GloMax-Multi Detection system (Promega). To correct for transfection efficiency, the luciferase activity was normalized to the renilla luciferase activity. Each experimental condition was assayed in triplicate.

## Histological and immunohistochemical analysis

Kidney samples were fixed in 4% neutral-buffered formalin before being embedded in paraffin. H&E and Sirius red were performed on 5 µm sections using standard procedures. Sections of 3 µm were deparaffinized for IHC, and antigen retrieval was performed with 10 mM citrate sodium buffer by using the PT Link (Dako). Endogenous peroxidase and nonspecific protein binding sites were blocked with 3% $H_2O_2$ and 4% BSA in 1X EnVision wash buffer (Dako), respectively. Incubation with the following primary antibodies: KIM1 (AF1817; R&D systems) and F4/80 (F4/80 [D2S9R] XP Rabbit mAb #70076; Cell Signaling) was carried out overnight at 4°C and, right afterwards, incubation with biotinylated goat anti-mouse or anti-rabbit IgG was performed for 1 h at 4°C. VECTASTAIN ABC Kit (Vector Laboratories) was used for detection of the biotinylated secondary antibodies. Tissue sections were revealed with 20 µg/ml 3,3'-DAB (Dako) and counterstained with hematoxylin. Images were taken at 10x magnification with a Nikon's Eclipse TE2000-U light microscope (Nikon Instrument Europe B.V). The intensities of Sirius red (collagen deposition) and IHC were quantified automatically with Image-Pro Plus software (Media Cybernetics). Data are represented as the percentage of the ratio of positive area to total tissue.

## Western blotting

Sections of kidney samples were homogenized and HPTEC were washed twice with cold PBS and lysed in RIPA buffer including 20 mM Tris–HCl, pH 7.5, 150 mM NaCl, 1% sodium deoxycholate, 1% NP-40, 0.1% SDS, protease inhibitors (Complete, Roche Diagnostics), and phosphatase inhibitors (Sigma-Aldrich). After normalizing for equal protein concentration, lysates were resuspended in SDS sample buffer before separation by SDS–PAGE and transferred onto nitrocellulose membranes. The membranes were incubated with the antibodies listed in Table S4. Images were acquired with the Odyssey Infrared Imaging System (LI-COR Biosciences), and densitometry was performed using the ImageJ 1.52p software (NIH).

## RNA isolation and qRT-PCR

The RNA from kidneys and the different cell cultures was isolated using a phenol–chloroform RNA extraction protocol. RNA samples were treated with RNase-Free DNase set (Cat 79254; QIAGEN) following manufacturer's instructions. iScript cDNA Synthesis kit (Bio-Rad) was used for the reverse transcription, following manufacturer's instructions. qRT-PCR was performed in triplicates using

the iQSYBR Green Supermix (Bio-Rad) and the primers described in Table S6. Relative mRNA expression was calculated using the ΔΔCt method (66), and the mRNA levels were normalized to 18S.

## TaqMan gene expression assay

Expression profile of a set of mouse genes related to fibrosis and metabolism was analyzed by using specific and unique TaqMan probes (Table S7). After the extraction of RNA, reverse transcription was carried out with the High-Capacity cDNA Reverse Transcription Kit (Thermo Fisher Scientific). RT–PCR was performed with the TaqMan Master (Thermo Fisher Scientific) in a Roche LightCycler 480 Real-Time PCR system (AB7900HT). Relative mRNA expression was calculated using the ΔΔCt method, and the mRNA levels were normalized to ubiquitin.

## Flow cytometry

Multiparametric flow cytometry was carried out for the identification of monocyte, macrophage, and neutrophil cell populations. To accomplish that, kidneys were collected from mice right after perfusion with 1X PBS. They were minced with a blade and incubated in agitation at 37°C for 20 min with 1 mg/ml of collagenase from Clostridium histolyticum (Sigma-Aldrich) in 1X PBS for tissue disaggregation. Samples were filtered (40 µm) and red blood cells were lysed by their incubation with RBC lysis buffer (BioLegend) for 1 min. Samples were washed twice, centrifuged for 10 min, 800$g$ at 4°C and cells were resuspended in 1x PBS. 2 × 10$^6$ cells were preincubated with CD16/CD32 (clone 2.4G2, BD, Bioscience) for 5 min at 4°C in 1X PBS. After two washing cycles with 1x PBS, cells were incubated with Ghost Dye Red 780 (Tonbo Biosciences) for Clock$^{\Delta 19/\Delta 19}$ mice experiments and Zombie violet (BD Biosciences) for CDKO experiment in protein/serum-free 1X PBS for 30 min at 4°C in darkness to stain dead cells. Cells were washed twice in FACS buffer containing 1% FBS and 0.5% BSA in 1x PBS and then incubated with the antibodies (BioLegend) listed in Table S8 in darkness for 1 h at 4°C. Antibody incubation was performed in Brilliant Stain buffer (BD Biosciences) for a better resolution of the brilliant violet–conjugated antibodies used in this study. Finally, cells were washed twice and fixed with 4% PFA in 1X PBS. Samples were analyzed on a FACSCanto II cytometer with DIVA software (BD Biosciences). Single stain and flow minus one controls for each fluorophore were used to compensate and establish gates, respectively. Data analysis was performed using FlowJo v10.6.2 software (FlowJo, LLC). Total cells were gated based on the forward versus side scatter (FSC versus SSC). Subsequently, single cells were selected by using forward height versus forward area (FSC-H versus FSC-A) plot, and dead cells were excluded with the corresponding viability marker described above. The identification of inflammatory and hematopoietic cells was based on the presence of CD45. Macrophages were identified by their expression of both F4/80 and CD11b markers. CD86 and CD206 markers were used to identify pro-inflammatory and anti-inflammatory macrophages, respectively. Pro-inflammatory monocytes and neutrophils were identified by their expression of CD11b and the absence of F4/80, and both populations were differentiated by their differential expression of Ly6C and Ly6G.

## Mitochondrial copy number determination

DNeasy Blood and Tissue Kit (Cat. 69504; QIAGEN) was used for the extraction of genomic DNA from kidneys according to the manufacturer's instructions. Mitochondrial DNA copy number was characterized with the Mouse Mitochondrial DNA Copy Number Assay Kit (Detroit R&D). Relative mtDNA copy number was represented as the mtDNA to nuclear DNA ratio.

## Soluble collagen quantification

5 mg of frozen kidneys were incubated overnight in 0.5 M acetic acid with 0.1 mg/ml pepsin (Sigma-Aldrich) at 4°C. Collagen quantification was performed with Soluble Collagen Assay Kit (Cat K532-100; BioVision) following manufacturer's instructions. Fluorescence was recorded using the ClarioStar plus (Bmg Labtech), and data were normalized with total protein measured with Pierce BCA protein assay kit (Cat 23227; Thermo Fisher Scientific).

## Statistical analysis

Data analysis was performed using GraphPad Prism 8.0 (GraphPad Software). Data are represented as mean ± SEM. Statistical differences were determined with the non-parametric Mann–Whitney test when two independent groups were analyzed, whereas one-way ANOVA was used for more than two independent groups and one independent variables and the non-parametric Wilcoxon test was used for paired analysis. A $P$-value of 0.05 was considered to be statistically significant.

## Ethics statement

The protocols herein described regarding animal experimental models and procedures were approved by local corresponding authorities, namely, the Comunidad de Madrid in Spain under PROEX 098.0/22 and the Ethical Committee of Salk Institute (California) in the USA.

## Supplementary Information

## Acknowledgements

This work was supported by grants from the Ministerio de Ciencia e Innovación PID2019-104233RB-100/AEI/10.13039/501100011033 (S Lamas), Instituto de Salud Carlos III REDinREN RD12/0021/0009 and RD16/0009/0016 (S Lamas), Comunidad de Madrid "NOVELREN" B2017/BMD-3751 and INNOREN P2022/BMD-7221 (S Lamas and C Barbas), and Fundación Renal "Iñigo Alvarez de Toledo" (S Lamas), all from Spain. C Rey-Serra has been the recipient of an FPI research training contract from the Spanish Research State Agency (BES-2016-076735). The CBMSO receives institutional support from Fundación "Ramón Areces." We acknowledge the laboratories of Fernando Rodríguez Pascual (CBMSO) for helping with plasmid constructions and of Marta Ruiz-Ortega at the Fundación Jiménez Díaz for helping with immunohistochemistry. We also acknowledge the help of the following facilities of the CBMSO: animal housing, flow cytometry, and confocal and electron microscopy.

## Author Contributions

C Rey-Serra: conceptualization, data curation, formal analysis, investigation, visualization, methodology, and writing—original draft, review, and editing.
J Tituaña, T Lin, JI Herrero, and V Miguel: investigation and methodology.
C Barbas: methodology.
A Meseguer: resources.
R Ramos: resources and software.
A Chaix: conceptualization, investigation, and methodology.
S Panda: resources and methodology.
S Lamas: conceptualization, data curation, formal analysis, supervision, funding acquisition, validation, investigation, visualization, project administration, and writing—original draft, review, and editing.

## Conflict of Interest Statement

The authors declare that they have no conflict of interest.

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
