## [Reviewer comments · Life Science Alliance]

Life Science Alliance

Reciprocal regulation between the molecular clock and kidney injury

Carlos Rey Serra, Jessica Tituaña, Terry Lin, Jose Herrero, Verónica Miguel, Coral Barbas, Anna Meseguer, Ricardo Ramos, Amandine Chaix, Satchidananda Panda, and Santiago Lamas

DOI: <https://doi.org/10.26508/lsa.202201886>

Corresponding author(s): *Santiago Lamas, Centro de Biología Molecular Severo Ochoa and Carlos Rey Serra,*

Review Timeline:	Submission Date:	2022-12-22
	Editorial Decision:	2023-02-10
	Revision Received:	2023-05-30
	Editorial Decision:	2023-06-07
	Revision Received:	2023-07-02
	Accepted:	2023-07-03

Transaction Report:

February 10, 2023

Re: Life Science Alliance manuscript #LSA-2022-01886-T

Prof. Santiago Lamas
Centro de Biología Molecular
Department of Cell Biology and Immunology
Nicolas Cabrera 1
Campus UAM
Madrid, Outside the United States or Canada 28049
Spain

Dear Dr. Lamas,

Thank you for submitting your manuscript entitled "Crosstalk between the molecular clock and kidney inflammatory, metabolic and fibrotic responses" to Life Science Alliance. The manuscript was assessed by expert reviewers, whose comments are appended to this letter. We invite you to submit a revised manuscript addressing the Reviewer comments.

Thank you for this interesting contribution to Life Science Alliance. We are looking forward to receiving your revised manuscript.

Sincerely,

B. MANUSCRIPT ORGANIZATION AND FORMATTING:

Reviewer #1 (Comments to the Authors (Required)):

Life Sciences review Jan 2023

Rey-Serra et al. present a plethora of data in this paper investigating the role of specific clock proteins in renal fibrosis. The main findings support a role for BMAL1, CLOCK, and CRY in the response to inflammation and fibrosis, although the timing and the setting (depending on model and length of renal injury) are variable. Loss of both CRY1 and CRY2 in mice resulted in worsened fibrosis and alterations to metabolic gene expression. In general, this manuscript is quite impressive and represents a tremendous amount of work. A major issue is that the data presentation takes away from the main message. This paper could easily be broken down into 2 or even 3 separate manuscripts. If that is not an option, additional descriptions of rationale, experimental design, as well as limitations would greatly improve the presentation of the manuscript.

Abstract

1. Suggest mentioning the fibrosis models used in the abstract

Introduction

2. MCP acronym for molecular clock pathway is not standard and should be removed. Likewise for CR to abbreviate circadian rhythm.

3. Gene names should be italicized

4. Please add the additional detail that the "master clock" is located in the SCN of the hypothalamus

5. Suggest adding additional details about the connections between the clock and metabolism.

6. The authors should consider two comprehensive reviews on the circadian clocks in the kidney (PMID: 31635525 and PMID: 35575250).

Methods

7. There is no statement regarding regulatory approval for these studies (IACUC or similar).

8. In general, the methods section is hard to follow. Suggest placing the animal methods at the start and presenting in order of the animal studies. Including an experimental design diagram would be helpful.

9. Suggest calling Cry1/Cry2 double KO by a different acronym as CDKO suggests a collecting duct knockout.

Results

10. Figure 1. Do the authors know what time of day the samples were collected for the database transcriptomic studies? The positive arm (BMAL1/CLOCK) of the clock peaks opposite that of the negative arm (CRY/PER) so that should be considered when interpreting the data. This reviewer suspects that detail was not included in the publication (it rarely is!), but in this case, it would be worth a personal communication to try to ascertain this key detail.

11. Figure 2. Did the authors consider that the clock is being activated by addition of TGFbeta? TGFbeta has been shown to reset the clock (PMID: 19029909; PMID: 29940038).

12. Figures 3 and 4. There is so much data presented, it is difficult to understand the overall message.

13. What is the rationale for the timing of tissue collection in the various models? Sometimes its 3 days, sometimes 5, one experiment goes 25 days. It is not clear how or why these time points were chosen.

14. Suggest caution in interpreting the lesser effect in the BMAL1 KO as only the earlier time points were tested.

Discussion

15. Not sure crosstalk is the most appropriate term. What the authors describe is a kind of reciprocal regulation.

16. The authors should discuss Ref 55 in more detail relative to the current findings, especially since the BMAL1 KO in the current study was a global KO and is known to have a number of different pathologies.

17. Ref 33 should also be taken into consideration, particularly as it is relevant to the TGF beta data

Reviewer #2 (Comments to the Authors (Required)):

In their study, Rey-Serra et al explore the relationship between the molecular circadian clock and the inflammatory and fibrotic responses in experimental renal injury models. Through the usage of several KO mouse lines the authors do find a relationship between circadian rhythm genes, especially Cry genes, and inflammation/fibrosis in renal injury models. The employed methodologies are well appropriate, the paper is data-rich and well written, while this is the first study to highlight the

involvement of the Cry genes in renal physiology.

Please find below several suggestions on how to improve the manuscript:

Please discuss/cite previous publications reporting the expression of clock genes in UOO and/or human renal fibrosis.

To possibly "correlate" the expression of Arntl with TGF in UOO, the appropriate statistical test should be used.

Please explain the selection of the particular scRNAseq dataset. Do other available datasets produce similar results? Please report possible differential expression of the (investigated in this manuscript) clock genes in the identified different cell populations (violin plots with statistics and not just visualization). The same is true for the inflammatory markers and the TGF pathway. Fig. 1 F,H,J, L, M should be moved to sup figures (or deleted); very low level info and not convincing.

Please elaborate on the TGFb-induced upregulation of clock genes. Have these results reported before for the kidney UOO/CKDs)? Which cell type? (cite/discuss)

In Figure 2O why are Smad3 levels lower in the presence of Tgfb+smad3pCMV5 as compared to smad3pCMV5 alone? Have the authors checked the protein levels of Smad3 upon its overexpression (e.g. with Western or immunofluorescence)? Proof should be provided that the transfection has worked. Did TGF phosphorylate the exogenous SMAD? Are the potential binding sites of Smad3/4 two as stated in the text or three as shown in the table (Fig. 1L)?

Pharmacologic inhibition with SB505124 ideally should have been performed with increasing concentrations. Was the viability of the cells affected (e.g. MTT)? How about Cry1&2 WB? An RNAi would have been more convincing to assign a role to the Alk 5 receptor. What was the diluent/vehicle of TGF/SB? Was it used in the control reaction? The western for Clock in 2E-F shows downregulation upon Tgfb which is in contrast to what is shown in 2B-C; please explain.

The order of presentation of the UOO results in the different KO strains is confusing. Moreover, procedures and read out assays should be identical. Please first show activity patterns, then prove the development, or not of fibrosis, to be followed by FACS and gene expression measurements. A comparative table would be helpful.

Why Bmal and clock deletion were assessed only at the acute phase of UOO? What happens at the peak of the UOO-induced disease? Moreover, as the Bmal1 KO mouse is a tamoxifen-inducible one, and given the tmx effects in inflammation, several additional control groups should have been included.

Metabolic aspects around the CR could be examined by further experimentation (such as exposure to high levels of triglycerides or cholesterol - in vitro mimicking the causes that promote CKD in humans) and examine the cellular response both in terms of CR aberrations and metabolic reprogramming, dissected in the pathways these authors suggest in the present expression profiling. The paper would greatly gain strength if the authors performed metabolic experiments such as measuring the fatty-acid oxidation rate or the ATP levels in kidneys from the Cry1+2 double KO mice (compared to WT mice) during UUO, so as to establish the relationship between the Circadian clock genes and the metabolic pathways.

In Figure 6B it doesn't seem that TFAM protein levels are lower in CDKO mice compared to single Cry1 and Cry2 null mice as written in the corresponding text. Are the proposed reductions significant in figure 6D?

Reviewer #1

Rey-Serra et al. present a plethora of data in this paper investigating the role of specific clock proteins in renal fibrosis. The main findings support a role for BMAL1, CLOCK, and CRY in the response to inflammation and fibrosis, although the timing and the setting (depending on model and length of renal injury) are variable. Loss of both CRY1 and CRY2 in mice resulted in worsened fibrosis and alterations to metabolic gene expression. In general, this manuscript is quite impressive and represents a tremendous amount of work. A major issue is that the data presentation takes away from the main message. This paper could easily be broken down into 2 or even 3 separate manuscripts. If that is not an option, additional descriptions of rationale, experimental design, as well as limitations would greatly improve the presentation of the manuscript.

We thank the reviewer for the praise to our manuscript. At this point we do not plan to break it down into separate manuscripts. We have attempted to improve the description of the rationale and the experimental design and to expand the discussion on the limitations.

Abstract

1. Suggest mentioning the fibrosis models used in the abstract

We have done as suggested.

Introduction

2. MCP acronym for molecular clock pathway is not standard and should be removed. Likewise for CR to abbreviate circadian rhythm.
3. Gene names should be italicized
4. Please add the additional detail that the "master clock" is located in the SCN of the hypothalamus
5. Suggest adding additional details about the connections between the clock and metabolism.
6. The authors should consider two comprehensive reviews on the circadian clocks in the kidney (PMID: 31635525 and PMID: 35575250).

Thank you for the suggestions. We have followed the reviewer's recommendations and incorporated them in the Introduction section.

Methods

7. There is no statement regarding regulatory approval for these studies (IACUC or similar).

We have included a paragraph detailing the Ethical approval for animal procedures.

8. In general, the methods section is hard to follow. Suggest placing the animal methods at the start and presenting in order of the animal studies. Including an experimental design diagram would be helpful.

We have attempted to follow the suggestion of the reviewer. Regarding the experimental design diagram, we believe that the schematics we show in all figures containing experimental designs with mice should suffice to understand the experimental protocols that we followed.

9. Suggest calling Cry1/Cry2 double KO by a different acronym as CDKO suggests a collecting duct knockout.

Although we understand the point of view of the reviewer, we are not aware of the existence of a “collecting duct knockout”. Thus, we have opted for keeping the acronym since its change would force a major formal renaming in many figure panels. Besides, the acronym CDKO has already been used in a previous publication from the group of Dr. Panda (co-author): please see Chaix et al, *Cell Metabolism* 29: 303-19, 2019, reference 13 of the manuscript. We hope to meet the indulgence of the reviewer regarding this issue.

Results

10. Figure 1. Do the authors know what time of day the samples were collected for the database transcriptomic studies? The positive arm (BMAL1/CLOCK) of the clock peaks opposite that of the negative arm (CRY/PER) so that should be considered when interpreting the data. This reviewer suspects that detail was not included in the publication (it rarely is!), but in this case, it would be worth a personal communication to try to ascertain this key detail.

Thanks for the suggestion. We checked with the laboratory that had generated the database and they confirmed that the mice were under a 12/12 light/dark cycle and that samples were collected between ZT2 and ZT5, this implying that it took place during the light period. We have now specified this in the Results section. As the reviewer points out and in consistence with published work (PMID 15780093 and 27056296), the expression of genes related to the positive arm of the circadian clock peaks at ZT0 so it should be declining only moderately at these time points. By contrast the expression of genes related to the negative arm peak at ZT12, far from the time points of the study. The fact that most of the differences encountered in this study were in genes related to the positive arm is therefore consistent.

11. Figure 2. Did the authors consider that the clock is being activated by addition of TGFbeta? TGFbeta has been shown to reset the clock (PMID: 19029909; PMID: 29940038).

We thank the reviewer for this interesting question. We believe that in Figure 2 we show that TGF-beta modifies key components of the molecular clock. In HPTEC in particular *Arntl/Bmal1* was the earliest component to become upregulated after exposure to TGF-beta. Thus, it is likely that the clock is being reset in these cells, though we did not perform a formal assessment of circadian phases in vivo. The two references alluded by the reviewer demonstrate resetting of the clock by TGF-beta in mammalian cells and zebrafish by either *Per1*-independent (mammalian cells) and dependent (zebrafish) mechanisms, a question which we did not address.

12. Figures 3 and 4. There is so much data presented, it is difficult to understand the overall message.

Thanks for the comment. Both figures have thematic consistence and we feel they can be left as they are without major modification. Regarding the overall message we have attempted to make it clearer in the Results section.

13. What is the rationale for the timing of tissue collection in the various models? Sometimes its 3 days, sometimes 5, one experiment goes 25 days. It is not clear how or why these time points were chosen.

We performed a time course in the UUO model (3, 7, 15 and 25 days) and we analyzed the expression of the clock-related genes 7 and 15 days after FAN and 25 days after ADN (Figures 1 and supplementary figure 1). In the UUO model we observed a higher expression of fibrosis-related genes than in the FAN and ADN models at their respective time points. Moreover, the UUO model was associated with significantly more collagen deposition than the FAN and ADN models: 15 days after FAN and 25 days after ADN showed a deposition of collagen equivalent to 3 and 7 days after UUO, respectively. Our rationale for the timing of the tissue collection in the different models was based on the differences in the progression of tissue damage present in the models analyzed. The observations mentioned above, together with an increased mortality related to FAN and ADN, strongly supported the use of UUO for further studies. This information has been included in the Results section.

14. Suggest caution in interpreting the lesser effect in the BMAL1 KO as only the earlier time points were tested.

Supplementary figure 4 shows data in the BMAL1 KO in the UUO model after 3 and 7 days, which correspond to inflammatory and pre-fibrotic changes, respectively. We have followed the reviewer's suggestion, the figure now includes the analysis of the mRNA expression of inflammatory markers and the F4/80 IHC 7 days after UUO. In the results section of the text, we state there were no significant changes respect to control in either condition, including expression of cytokines and inflammatory markers. Following, the suggestion of the reviewer we have added a sentence of caution in the Discussion section.

Discussion

15. Not sure crosstalk is the most appropriate term. What the authors describe is a kind of reciprocal regulation.

We had been thinking about this for a long time. Each expression has its highlights and downsides. Beyond entering a discussion into the semantics, we have decided to adhere to the reviewer's comment and change the title accordingly.

16. The authors should discuss Ref 55 in more detail relative to the current findings, especially since the BMAL1 KO in the current study was a global KO and is known to have a number of different pathologies.

17. Ref 33 should also be taken into consideration, particularly as it is relevant to the TGF beta data

Thank you for the suggestions. We have now discussed the mentioned references in detail now ref 39 and 34, respectively, together with a very recent publication showing interesting results in an inducible model of Bmal1 deletion in the renal tubule (Bignon et al, Ref. 64).

Reviewer #2 (Comments to the Authors (Required)):

In their study, Rey-Serra et al explore the relationship between the molecular circadian clock and the inflammatory and fibrotic responses in experimental renal injury models. Through the usage of several KO mouse lines the authors do find a relationship between circadian rhythm genes, especially Cry genes, and inflammation/fibrosis in renal injury models. The employed methodologies are well appropriate, the paper is data-rich and well written, while this is the first study to highlight the involvement of the Cry genes in renal physiology.

We thank the reviewer for the laudatory comments.

Suggestions:

1. To possibly "correlate" the expression of Arntl with TGF in UOO, the appropriate statistical test should be used.

We believe the reviewer is alluding to Fig. 1B. We were not sure about the statistical test to which the reviewer is referring. However, we attempted a Spearman correlation analysis of the heatmap intensities and we found statistically significant correlations between circadian genes (Arntl, Arntl2, Cry1, Clock, Npas2, per1 and Nr1f1) and fibrosis-related genes (Tgfb and Fn1). The Spearman and p-value are shown in figures 1C and Sup Table 1: In addition, we also performed the Spearman Correlation analysis in the FAN and ADN models with data represented in Sup Fig 1A, B. The new data are shown in Sup Figs 1C, D.

2. Please explain the selection of the particular scRNAseq dataset. Do other available datasets produce similar results?

This scRNAseq data base was selected because the mouse model employed was similar to ours in terms of age (8-12 weeks), CKD model (UUO) and genotype (C57BL6). This is now specified in the Results section. The data were obtained from the Ref 21.

Following the reviewer's suggestion we also analyzed a SC dataset from a model of ischemia-reperfusion injury from the Ref 22 and obtained comparable results. This is now specified in the Results Section.

3. Please report possible differential expression of the (investigated in this manuscript) clock genes in the identified different cell populations (violin plots with statistics and not just visualization). The same is true for the inflammatory markers and the TGF pathway.

We have now included the violin plots from the expression of circadian genes in the different cell populations. This is now included in Figure 1H, K. With respect to inflammatory markers and TGF beta genes we have decided to leave out those results as their relevance was only minor. the figures related to that results.

4. Fig. 1 F,H,J, L, M should be moved to sup figures (or deleted); very low level info and not convincing.

In keeping with the reviewer's suggestion, we have re-organized the figure and kept in the main figures only the panels we believe are important to convey the main message. The rest have been left out.

5. Please elaborate on the TGFb-induced upregulation of clock genes. Have these results reported before for the kidney UOO/CKDs)? Which cell type? (cite/discuss).

As we mention in the discussion, most of the published literature has addressed the influence of circadian rhythm and clock genes on TGF-beta (Refs 36-39). As described in Ref.# 37, TGF-b is able to increase Bmal1 expression in lung epithelial cells and fibroblasts. However, there is little previous evidence showing that TGF-beta upregulates clock genes in the kidney, in particular Bmal1. Overall, the role of Bmal1 in kidney fibrosis is still unclear. The postnatal suppression of whole body Bmal1 has been shown to be protective against fibrosis in the UUO model and these same authors also found upregulation of murine tubular cells treated with TGF-beta (Zhang et al, ref 39). However, Liu et al (ref 57) showed that proximal tubular Bmal1 deletion negatively affected the outcome of CKD in the Adenine-induced nephrotoxicity model. Of interest, recent work has shown that conditional deletion of Bmal1 in the renal tubule was associated with severe metabolic derangement (Firsov, JCI new ref 64), although fibrosis was not addressed in this work. With respect to our data, we found a clear upregulation of Bmal1 by TGF-beta in the UUO model as well as in PTECs. However, neither the global deletion of Bmal1 (this study) or our data in the conditional one in the proximal tubule (Rey-Serra et al, BioRxiv <https://doi.org/10.1101/2022.05.18.492458>) were consistent with a conspicuous fibrotic phenotype under the UUO model. It is possible that TGF-beta induced upregulation of Bmal1 could represent a compensatory mechanism by which cells attempt to counteract damage, although this is mere speculation. We have attempted to summarize all this in the Discussion section.

6. In Figure 2O why are Smad3 levels lower in the presence of Tgfb+smad3pCMV5 as compared to smad3pCMV5 alone? Have the authors checked the protein levels of Smad3 upon its overexpression (e.g. with Western or immunofluorescence)? Proof should be provided that the transfection has worked. Did TGF phosphorylate the exogenous SMAD? Are the potential

binding sites of Smad3/4 two as stated in the text or three as shown in the table (Fig. 1L)?

Thank you for the questions. First, we do not think that Smad3 levels in the presence of the plasmid pCMV5 alone (Fig2O, column 1) were significantly higher than Smad3 levels after Smad3 overexpression in the presence of TGF-beta (Fig2O, column 5). However, the reviewer poses a valid point concerning the protein levels after Smad3 overexpression. To address it, we have performed new transfection experiments that show conspicuous overexpression of Smad3 by immunoblot, as well increased phosphorylation of Smad3 (Fig2 P, Q). There are two binding sites for Smad3/4, the table shows three because JASPAR recognized the negative and positive strand of one of the two sites.

7. Pharmacologic inhibition with SB505124 ideally should have been performed with increasing concentrations. Was the viability of the cells affected (e.g. MTT)? How about Cry1&2 WB? An RNAi would have been more convincing to assign a role to the Alk 5 receptor. What was the diluent/vehicle of TGF/SB? Was it used in the control reaction? The western for Clock in 2E-F shows downregulation upon Tgfb which is in contrast to what is shown in 2B-C; please explain.

Thank you for the questions. We agree with the reviewer about the logic of increasing concentrations of the TGF-beta signaling inhibitor, however our aim was only to prove specificity of the signaling pathway. For the sake of viability, we did an MTT assay with concentrations of the inhibitor between 0.5 and 3.5 uM and found no significant toxicity (Fig. S3A). This is now reported in the Results section and the MTT assay has been included in the methods section.

We have attempted to do Western blots to explore differences in Cry1/2 protein levels after TGF-beta treatment in the presence and absence of the inhibitor. However, the antibodies we tried failed to provide an interpretable signal. Moreover, in our hands, as shown in Fig.1A only Cry1 mRNA was regulated by TGF-beta. Regarding the suggestion of the reviewer on the use of RNAi for Alk5, again our aim was to prove the specificity of the pathway rather than to explore the receptor involved. Even though Alk5 is the TGFb1 receptor that is mainly inhibited by SB505124, we decided to remove the mention to Alk5 from the manuscript. All the control conditions were in the presence of the appropriate diluent/vehicle as described in the Methods section. Regarding the downregulation of Clock levels shown in Fig.2D, E (non-significant), we decided to repeat those experiments and we did not observe significant increase or decrease of its expression. This can be now seen in Fig. 2D, E. The differences observed in terms of Clock expression between Fig. 2A-C and Fig. 2D, E might be attributed to the different time of exposure to TGF-beta, which was 24 h in this case, compared to the 0-12 h time course of Fig.2B-C.

8. The order of presentation of the UOO results in the different KO strains is confusing. Moreover, procedures and read out assays should be identical. Please first show activity patterns, then prove the development, or not of

fibrosis, to be followed by FACS and gene expression measurements. A comparative table would be helpful.

Thanks for the suggestion. We have attempted to present the UUO results in a clearer manner. We have followed the reviewer's suggestions, so we first show fibrosis and then inflammation. However, activity patterns were only measured in the Bmal1 KO since it was the only model that was conditional.

9. Why Bmal and clock deletion were assessed only at the acute phase of UOO? What happens at the peak of the UOO-induced disease? Moreover, as the Bmal1 KO mouse is a tamoxifen-inducible one, and given the tmx effects in inflammation, several additional control groups should have been included.

Bmal and Clock deletions were assessed at 3 days and 7 days, which we believe are representative for the inflammatory phase and early fibrotic phase, respectively. Regarding the important question on potential interference of Tamoxifen in inflammation, it is important to remember that, as stated in Methods, mice were treated daily with 2 mg/ml of doxycycline in drinking water or 3.7 mg of tamoxifen administered by oral gavage for 2 weeks or 5 days, respectively. Subsequent experiments were performed four or two weeks after the end of the treatment. Nevertheless, we studied the staining of F4/80 in the presence and absence of Tamoxifen with the UUO model and found that in the absence of Tamoxifen, inflammation was clearly present after 7 days of UUO as shown in the graph below:

F4/80 IHC: #P<0.05 compared to their corresponding contralateral

qPCR data: #P<0.05 compared to their corresponding contralateral, *P<0.05 compared to the Bmal1 KO UUO with Tamoxifen, \$P<0.05 compared to the WT UUO with Tamoxifen

3 Days after UUO

7 Days after UUO

As shown above, we also analyzed by qPCR the levels of inflammatory markers after 3 and 7 days of UUO in the Bmal1 (F4/80, IL-6, TNF-a, CD80, CD86 and IFN-gamma) and Clock KOs (CD206, IL4 and IL10) and found no significant effects of Tamoxifen in terms of decreasing the levels of these mediators after UUO, with the exception of IL-6. Thus, we believe that these data do not provide a basis for the reinterpretation of our results and we added a sentence to make this clear in the Methods section.

10. Metabolic aspects around the CR could be examined by further experimentation (such as exposure to high levels of triglycerides or cholesterol - in vitro mimicking the causes that promote CKD in humans) and examine the cellular response both in terms of CR aberrations and metabolic reprogramming, dissected in the pathways these authors suggest in the present expression profiling. The paper would greatly gain strength if the authors performed metabolic experiments such as measuring the fatty-acid oxidation rate or the ATP levels in kidneys from the Cry1+2 double KO mice (compared to WT mice) during UUO, so as to establish the relationship between the Circadian clock genes and the metabolic pathways.

We certainly agree with the reviewer about the interest and importance of exploring the metabolic changes associated to circadian rhythm dysfunction in a more profound way. However, due to lack of availability of the Cry1/2 mice we are not in a position to perform these experiments, which in any event should be part of a separate manuscript. As mentioned above, some of the functional

changes in kidney metabolism have just been reported in the conditional Bmal1 KO (Firsov JCI, new ref 64).

11. In Figure 6B it doesn't seem that TFAM protein levels are lower in CDKO mice compared to single Cry1 and Cry2 null mice as written in the corresponding text. Are the proposed reductions significant in figure 6D?

Thank you for the observation. The reviewer is right. We have rewritten the text to make it clearer.

June 7, 2023

RE: Life Science Alliance Manuscript #LSA-2022-01886-TR

Prof. Santiago Lamas
Centro de Biología Molecular Severo Ochoa
Department of Cell Biology and Immunology
Nicolas Cabrera 1
Campus UAM
Madrid, Outside the United States or Canada 28049
Spain

Dear Dr. Lamas,

Thank you for submitting your revised manuscript entitled "Reciprocal regulation between the molecular clock and kidney injury". We would be happy to publish your paper in Life Science Alliance pending final revisions necessary to meet our formatting guidelines.

- please upload your Tables in editable .doc or Excel format
- please add your main, supplementary figure, and table legends to the main manuscript text after the references section
- please add ORCID ID for the secondary corresponding author--they should have received instructions on how to do so
- please be sure that the Authors' names match in the system and on the manuscript file (Jessica Paola Tituaña-Fajardo in the system vs. Jessica Tituaña in the manuscript)
- please consult our manuscript preparation guidelines <https://www.life-science-alliance.org/manuscript-prep> and make sure your manuscript sections are in the correct order
- please add an Author Contributions section to your main manuscript text
- please indicate the scale bar size in Legend for Fig. 2G
- please add scale bars to the figure S5 and S6 accordingly
- it is hard to see the scale bars in figures 3B, 5E and S1F
- please indicate the molecular weight next to each protein blot
- please add a callout for Figure 5L to your main manuscript text
- please include callouts for all panels in Figure S5 in your main manuscript text.

A. FINAL FILES:

B. MANUSCRIPT ORGANIZATION AND FORMATTING:

Sincerely,

Reviewer #2 (Comments to the Authors (Required)):

The authors have satisfactorily addressed most of my concerns and suggestions.

July 3, 2023

RE: Life Science Alliance Manuscript #LSA-2022-01886-TRR

Prof. Santiago Lamas
Centro de Biología Molecular Severo Ochoa
Department of Cell Biology and Immunology
Nicolas Cabrera 1
Campus UAM
Madrid, Outside the United States or Canada 28049
Spain

Dear Dr. Lamas,

Thank you for submitting your Research Article entitled "Reciprocal regulation between the molecular clock and kidney injury". It is a pleasure to let you know that your manuscript is now accepted for publication in Life Science Alliance. Congratulations on this interesting work.

DISTRIBUTION OF MATERIALS:

Again, congratulations on a very nice paper. I hope you found the review process to be constructive and are pleased with how the manuscript was handled editorially. We look forward to future exciting submissions from your lab.

Sincerely,
